# COVID-19 cluster size and transmission rates in schools from crowdsourced case reports

Paul Tupper[1]\*, Shraddha Pai[2,3], COVID Schools Canada, Caroline Colijn[1]\*

[1]Department of Mathematics, Simon Fraser University, Burnaby, Canada; [2]The Donnelly Centre, University of Toronto, Toronto, Canada; [3]Ontario Institute for Cancer Research, Toronto, Canada

**Abstract** The role of schools in the spread of SARS-CoV-2 is controversial, with some claiming they are an important driver of the pandemic and others arguing that transmission in schools is negligible. School cluster reports that have been collected in various jurisdictions are a source of data about transmission in schools. These reports consist of the name of a school, a date, and the number of students known to be infected. We provide a simple model for the frequency and size of clusters in this data, based on random arrivals of index cases at schools who then infect their classmates with a highly variable rate, fitting the overdispersion evident in the data. We fit our model to reports from four Canadian provinces, providing estimates of mean and dispersion for cluster size, as well as the distribution of the instantaneous transmission parameter $\beta$, whilst factoring in imperfect ascertainment. According to our model with parameters estimated from the data, in all four provinces (i) more than 65% of non-index cases occur in the 20% largest clusters, and (ii) reducing instantaneous transmission rate and the number of contacts a student has at any given time are effective in reducing the total number of cases, whereas strict bubbling (keeping contacts consistent over time) does not contribute much to reduce cluster sizes. We predict strict bubbling to be more valuable in scenarios with substantially higher transmission rates.

**\*For correspondence:**
pft3@sfu.ca (PT);
ccolijn@sfu.ca (CC)

**Group author details:**
COVID Schools Canada See page 12

## Editor's evaluation

This paper provides an important novel methodology to understand the mode of spread of SARS-CoV-2 in schools given sparse data.

## Introduction

In the management of the COVID-19 pandemic, an important consideration is the role of children and in particular schools. In most jurisdictions rates of SARS-CoV-2 infection among children are similar to those in the adult population (*Centers for Disease Control and Prevention, 2021*). But severity is much lower in children; the infection fatality rate (IFR) of COVID for at age 10 was estimated to be 0.002% versus an IFR of 0.01% at age 25, and 0.4% at age 55, for the original SARS-CoV-2 virus present in 2020 (*Levin et al., 2020*). Cases are more often asymptomatic among children, less likely to require hospitalization and ICU care (*Centers for Disease Control and Prevention, 2021*), and less likely to be classified as long COVID (*Sudre et al., 2021*). On the other hand, MIS-C is a serious condition sometimes resulting from SARS-CoV-2 infection (*CDC, 2021a*), and myocarditis happens more frequently as a side effect of infection among younger individuals (*Singer et al., 2022*).

Jurisdictions have had to make a choice between closing schools, with all the attendant social, economic, and psychological costs (*Chaabane et al., 2021*), and leaving schools open, allowing

**eLife digest** During the COVID-19 pandemic, public health officials promoted social distancing as a way to reduce SARS-CoV-2 transmission. The goal of social distancing is to reduce the number, proximity, and duration of face-to-face interactions between people. To achieve this, people shifted many activities online or canceled events outright. In education, some schools closed and shifted to online learning, while others continued classes in person with safety precautions.

Better information about SARS-CoV-2 transmission in schools could help public health officials to make decisions of what activities to keep in person and when to suspend classes. If safety measures lower transmission in schools considerably, then closing schools may not be worth online education's social, educational, and economic costs. However, if transmission of SARS-CoV-2 in schools remains high despite measures, closing schools may be essential, despite the costs.

Tupper et al. used data about COVID-19 cases in children attending in-person school in four Canadian provinces between 2020 and 2021 to fit a computer model of school transmission. On average, their analysis shows that one infected person in a school leads to between one and two further cases. Most of the time, no more students are infected, indicating that normally infection clusters are small; and only rarely does one infected person set off a large outbreak. The model also showed that measures to reduce transmission, like masking or small class sizes, were more effective than interventions such as keeping students with the same cohort all day (bubbling).

Tupper et al. caution that their findings apply to the variants of SARS-CoV-2 circulating in Canada during the 2020-2021 school year, and may not apply to newer, highly transmissible strains like Omicron. However, the model could always be adapted to assess school or workplace transmission of more recent strains of SARS-CoV-2, and more generally of other diseases. Thus, Tupper et al. provide a new approach to estimating the rate of disease transmission and comparing the impact of different prevention strategies.

possible transmission of SARS-CoV-2 in that setting (*Centers for Disease Control and Prevention, 2021*). The direct downside of transmission in schools if it occurs is that children may be infected there, risking the low but non-negligible harms of COVID-19 in that age range, but also adult teachers and staff are put at risk. Transmission in schools may also contribute to overall community transmission, indirectly jeopardizing more vulnerable individuals (*Walsh et al., 2021*). As a concrete example, if a child contracts SARS-CoV-2 at school, they may then go on to transmit it to an elderly relative they live with, for whom the consequences are more severe (*Laws et al., 2021*). Estimating the magnitude of these two kinds of harm and making the decision as to what choice to make involves many sources of uncertainty and value judgements, which helps explain why different jurisdictions have taken different approaches (*Harris, 2020*). In some jurisdictions schools were open for the 2020–2021 school year, though many measures were put into place in order to reduce the risk of SARS-CoV-2 transmission (*British Columbia Ministry of Education, 2020*). Measures included cohorting, staggered entrance and exit times, masks, improvements in ventilation, extra sanitization measures. In other jurisdictions schools were closed for large portions of the year (*Partners, 2021*).

Studies that have looked at the effect of school closures on the overall rate of SARS-CoV-2 transmission find mixed results: some find substantial reduction in community transmission when schools are closed, and others small or no effect (*Walsh et al., 2021*; *Chernozhukov et al., 2021*). Given that schools involve many children all sharing a room for many hours a day, it may be surprising that there is not a clearer evidence of significant transmission in schools. One explanation is that children may be less likely to transmit SARS-CoV-2 to each other, either by being less infectious or by being less susceptible (*Dattner et al., 2021*; *Viner et al., 2021*). But transmission in schools does occur, and it's worthwhile to estimate the magnitude and characterize the variation in it.

One source of evidence for transmission in schools are school exposure reports. Throughout the pandemic organizations have collected data submitted by volunteers about COVID cases in schools, and such data has subsequently been published online (*National Education Association, 2020*; *Covid Schools Canada, 2021*; *Support Our Students Alberta, 2022*). Data consists of reports of exposures or clusters in schools, either submitted by parents or determined from reading newspaper reports. Several such websites exist, though many ceased due to excessive workload after the 2020–2021

school year. In some jurisdictions there are also similar sources of data provided by local government (*Government of Ontario, 2021*; *State of Michigan, 2021*) or Public Health Agencies (*Vancouver Coastal Health, 2021*; *Health, 2021*).

Here, we propose a simple model of transmission in schools, and we use these data on cluster sizes to estimate parameters of the model for four Canadian provinces. Our model allows for heterogeneity in transmission rate, which is able to capture the considerable variability in the sizes of the clusters, with most exposures leading to no further cases (and so a cluster of size 1) but with few having a large number of cases (*Tufekci, 2020*). We estimate the mean and overdispersion parameters for different jurisdictions. We then use our parameter estimates in a couple of ways: firstly, we explore the overdispersion of cluster sizes in different jurisdictions, giving estimates of what fraction of all cases are in the 20% largest of all clusters. Secondly, we can obtain an estimate of the distribution of the transmission rate $\beta$, the rate at which a single infected individual infects a susceptible person when they are in contact. This parameter, in turn, could be used to simulate school transmission and explore the impacts of interventions (*Tupper et al., 2020*) as we explore for some parameter choices. In Appendix 1 we perform a similar analysis for eight US states, where only substantially less complete datasets were available.

Finally, two important changes have occurred in 2021 that we expect to impact cluster sizes in schools. On the one hand, in many jurisdictions, large portions of children aged 5 and up have been vaccinated with the Pfizer/BioNTech vaccine (*The New York Times, 2020*). According to the extent to which the vaccine protects against infection, we expect cluster size will be reduced, as fewer students will be infected if they have been immunized. Observed cluster size may be reduced further even than this, if the vaccine allows harder-to-detect infections to occur. On the other hand, now more infectious variants of the coronavirus have emerged; the Alpha, Delta, and Omicron variants have all had a higher estimated transmissibility than their predecessors (*CDC, 2021b*; *CDC, 2022*). Increased transmissibility would suggest larger cluster sizes, certainly among unvaccinated ages, but the relative impact of vaccination and the new variants together is difficult to gauge. Furthermore, changes in vaccination, transmission, and immune evasion may all lead to a change in the variability in cluster sizes.

## Materials and methods

Our data consist of reports of confirmed cases among students, teachers, and staff in schools in four Canadian provinces during the 2020–2021 school year. Data was collected by Dr Shraddha Pai with COVID Schools Canada (*Covid Schools Canada, 2021*), an initiative of the group Masks for Canada (*Canadian Doctors, Professionals, & Citizens for Masks, 2021*). We included the four provinces from this dataset with the most schools reporting cases with date information. For each school, there is a list of confirmed cases among students, teachers, and staff, along with the dates on which the cases were reported. We then assigned cases to clusters based on being at the same school and being reported within 7 days of each other; if the difference in date between two cases was less than or equal to 7 days, or they could be linked by a sequence of such cases, they were put in the same cluster. We chose 7 days on the basis of estimates of the serial interval for COVID-19 of approximately 5 days (*Rai et al., 2021*). (We explore other choices of window in Appendix 1.) Information was not available about whether the cases at the same school were in the same classroom. Accordingly, we interpret clusters as capturing all linked cases at a given school, and not just one classroom.

There is substantial uncertainty in whether each of our determined clusters of cases accurately represents a set of cases linked by transmission. For any cluster of two or more cases, it may be that two independent sets of cases are incorrectly included in the same cluster. This may lead us to overestimate the size of clusters. Likewise, any two of our clusters in the the same school that occur further apart than 7 days may in fact be linked by a chain of undetected transmission, leading to an underestimate of cluster size. Both these factors may occur in our data, but we neglect both of them, taking the observed cluster size as given by our method. We are also unable to distinguish between transmission occurring in a school and in social activities with classmates outside of school.

In a given jurisdiction, we assume exposure events occur according to a Poisson process with variable rate. Independently of this process, once an exposure event occurs at a school, we say $Z$ additional people are infected by the index case, for a total of $Z+1$ individuals in the cluster. The variable $Z$ includes individuals directly infected by the index case, as well as any subsequent infected

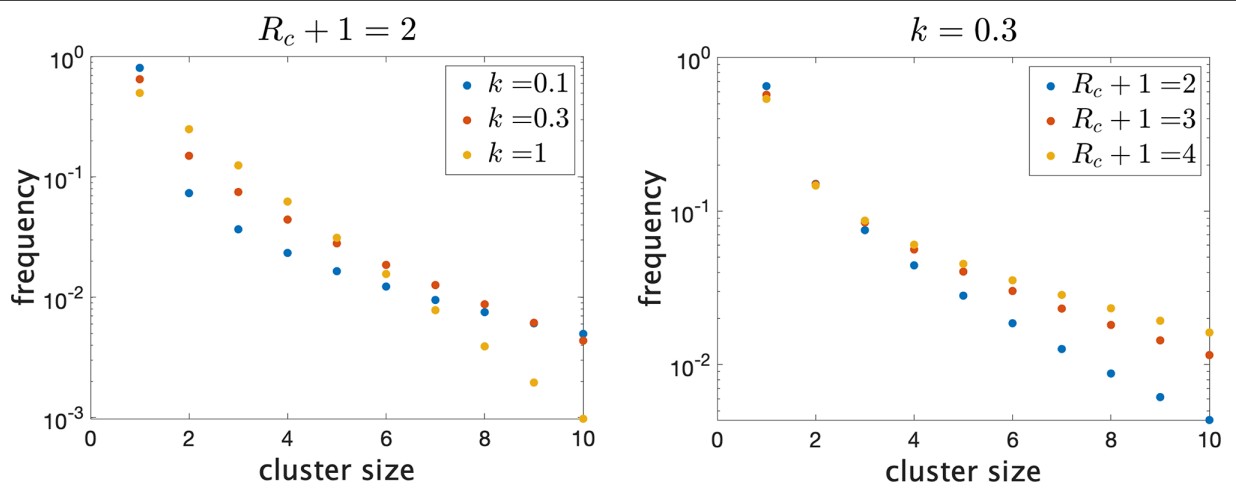

**Figure 1.** Frequency of clusters of different sizes on a log scale. Trends continue as shown for larger clusters. (Left) Fixing mean cluster size $R_c + 1$ and varying dispersion $k$. (Right) Fixing $k$ and varying $R_c + 1$.

individuals that are included in the same cluster. Following **Lloyd-Smith et al., 2005**, we model $Z$ as a Poisson random variable with parameter $\nu$, where $\nu$ itself is a Gamma-distributed random variable. As described by **Lloyd-Smith et al., 2005**, $Z$ is then a negative binomial random variable. Rather than the usual parametrization of a negative binomial distribution, we use parameters $R_c$ and $k$. The parameter $R_c$ is the expected number of additional infections in a cluster, and $k$ is the dispersion: a measure of how far the distribution of $Z$ is from being Poisson. As $k \to \infty$, the distribution of $Z$ approaches that of a Poisson distribution with mean $R_c$. The variance of $Z$ is $R_c(1 + R_c/k)$ and so for smaller values of $k$ we expect more of the secondary cases to occur in rare large clusters rather than in frequent small clusters (**Lloyd-Smith et al., 2005**).

There are then a total of $Z + 1$ infected individuals in the school. To give an idea of how the distribution of true cluster size depends on the parameters when they are in this range, in **Figure 1** we show the theoretical distributions for varying parameters. On the left, we fix $R_c + 1 = 2$ and vary $k$. Decreasing $k$ causes there to be more clusters of size 1 (i.e. no transmission) and more large clusters, but reduces the number of intermediate-sized clusters. On the right, we fix $k = 0.3$ and show the effect of varying mean cluster size $R_c + 1$. As $R_c$ increases, the frequency of clusters with no or little transmission decreases and the frequency of larger cluster sizes increases.

The number of the total $Z + 1$ cases that are actually observed, $X$, depends on the ascertainment model. We consider a model where each case is observed and contributes to the reported cluster size with probability q, so that the observed cluster size $X$ (conditioned on $Z$) is binomial with parameters $n = Z + 1$ and probability $q$. The index case is treated the same as the infectees, so $X$ may or may not include the index case. If none of the cases in a cluster are observed, we assume the cluster is not reported, so our model factors in the effect that smaller clusters are more likely to be missed. See Appendix 1 for an explicit statement of the likelihood function.

For each collection of cluster sizes in our datasets we estimate the mean $R_c$ and dispersion $k$ using the ascertainment model with q = 0.75. We base this value on the meta-analysis (**Bobrovitz et al., 2021**) which reports ascertainment fractions for high-income regions in the Americas between 66% (in the last quarter of 2020) and 85% (in the second quarter of 2021). We use maximum likelihood estimation to obtain estimates of $R_c$ and $k$, and we use the Hessian of the log-likelihood to obtain 95% confidence ellipses for the parameters [**Wasserman, 2013**, Sec. 9.10].

Finally, we perform a second analysis using the same model, using a smaller window of time for the definition of a cluster. In this way we hope to identify only the index case and the cases *directly* infected by the index case. We use the model above for this (smaller) number of cases for each cluster to estimate a distribution for $\nu$, but then use this in turn to estimate a distribution for the instantaneous transmission rate $\beta$. Our reasoning is that if $\nu$ is the random Poisson parameter when the index case it exposed to $n$ people for time $T$, then $\beta$ has approximately the same distribution as $\nu/(nT)$.

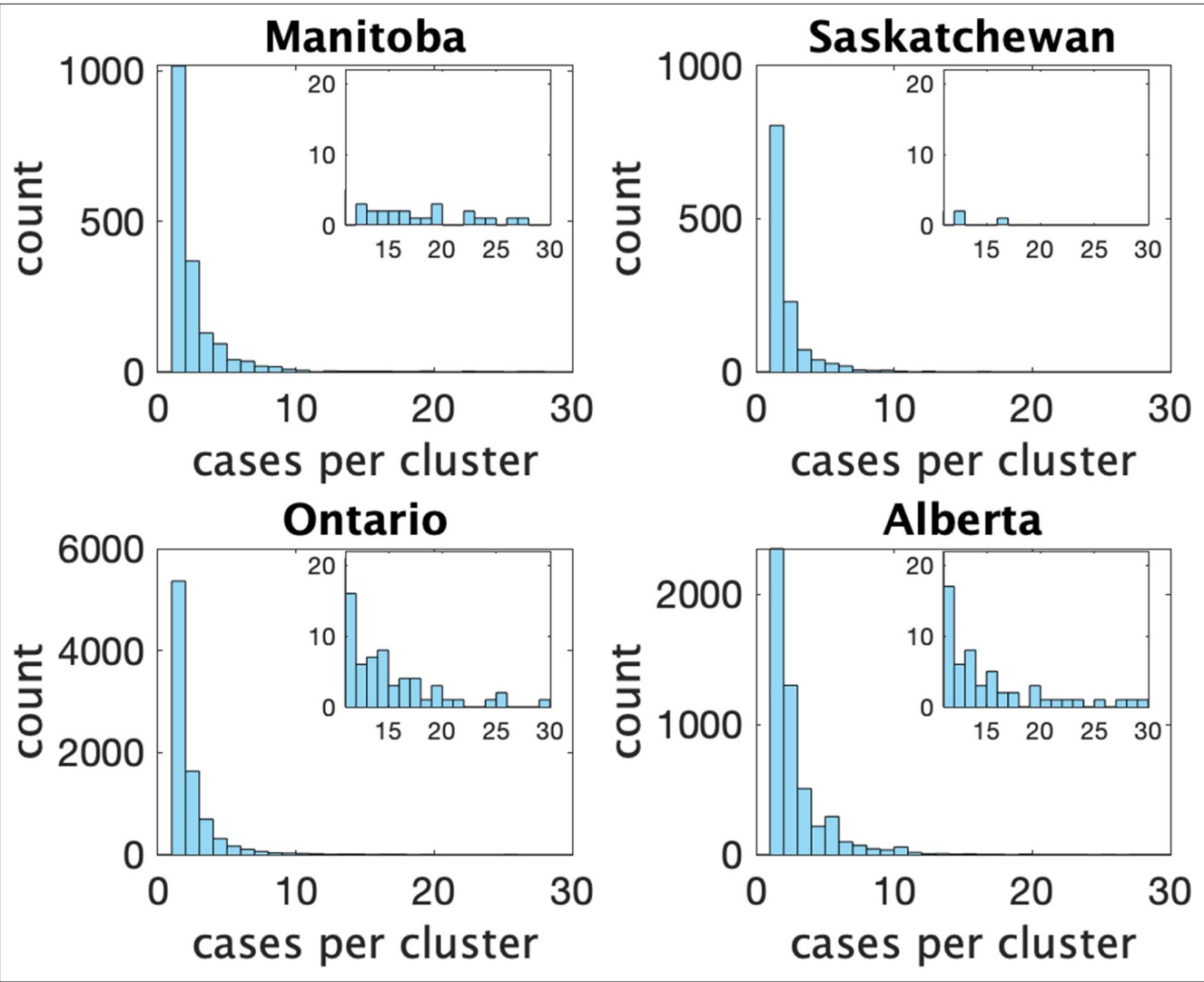

**Figure 2.** Histograms of observed cluster sizes in four Canadian provinces. Inset histograms only show clusters of size 11 or more on a different scale. Each dot represents a single cluster of size 11 or larger, and indicates the presence of (more rare) larger clusters.

Under these assumptions, $\beta$ is also a Gamma-distributed random variable with parameter we can easily identify, from those for $\nu$.

## Results

*Figure 2* shows histograms of cluster size according to our definition in the four provinces. In *Table 1* we show some statistics associated with the data for each province. In the top we show the number of clusters, the number of schools appearing, the number of schools with more than one reported cluster, and the fraction of schools with multiple clusters. In the bottom we show the fraction of clusters that have only one observed case, and the average number of observed cases in the clusters, the maximum observed cluster size, the index of dispersion (variance divided by mean) of cluster size, and index of dispersion of the number of cases in a cluster subtracting one for the presumed index case.

In *Figure 3* (left) we show the rate (in clusters per day per 100,000 population) that cases appear in the dataset over time. In *Figure 3* (right) we show the rate of COVID incidence per 100,000 population in the province over the same period of time. There is an apparent correspondence between the two time series, with peaks in rate of clusters per day occuring near peaks in incidence.

*Figure 4* (left) shows the estimated mean cluster size (= $R_c + 1$) and dispersion $k$ for the four Canadian provinces. Mean cluster sizes ranged from 1.9 to 2.9 cases, and dispersion ranged from 0.34 to

**Table 1.** Cluster statistics for each province.

(Top) For each of the four Canadian provinces: number of clusters in the data, number of schools reported, number of schools with multiple clusters, fraction of schools with multiple clusters. (Bottom) Fraction of clusters with one case, mean observed cluster size, maximum observed cluster size, and index of dispersion (variance of number of cases divided by mean number of cases) with and without subtracting one for the index case.

| Province | Number of Clusters | Number of Schools | Schools with Multiple clusters | Fraction of schools Multiple clusters |
|---|---|---|---|---|
| Manitoba | 1754 | 542 | 396 | 0.73 |
| Saskatchewan | 1211 | 466 | 295 | 0.63 |
| Ontario | 8482 | 3337 | 2147 | 0.64 |
| Alberta | 5032 | 1537 | 1158 | 0.75 |

| Province | Fraction with One case | Mean observed Cluster size | Max observed Cluster size | Index of Dispersion (IoD) | IoD without Index case |
|---|---|---|---|---|---|
| Manitoba | 0.58 | 2.16 | 44 | 3.44 | 6.43 |
| Saskatchewan | 0.66 | 1.70 | 16 | 1.23 | 2.98 |
| Ontario | 0.63 | 1.83 | 50 | 1.87 | 4.13 |
| Alberta | 0.47 | 2.45 | 108 | 4.94 | 8.35 |

0.53 (recalling that no overdispersion corresponds to $k \to \infty$.) Recall that we determined clusters by including cases in the same cluster if they were reported within 7 days of each other. In Appendix 1 we explore what happens if we change this window to either 4 or 10 days. We find that estimates of $k$ do not change much: there is less than a 10% change in $k$ in all cases. A window of 4 days leads to smaller cluster sizes (at most 18% smaller) and a window of 10 days leads to larger cluster sizes (at most 11% larger).

In Appendix 1 we explore varying the ascertainment fraction between 0.2 and 1. Though lower ascertainment fractions yield bigger values of $R_c$ and smaller values of $k$, we see that the parameter estimates are relatively insensitive to values of q between 0.5 and 1. For example, when $q_1$ is reduced from 0.75 to 0.5, the range of $R_c + 1$ shifts from 1.9–2.9 to 3.2–6.4, and the range of $k$ shifts from 0.34–0.53 to 0.22–0.39. The reason for this is that though a given cluster with multiple cases will look smaller with fewer cases detected, and lower detection will thereby bias observed size downwards,

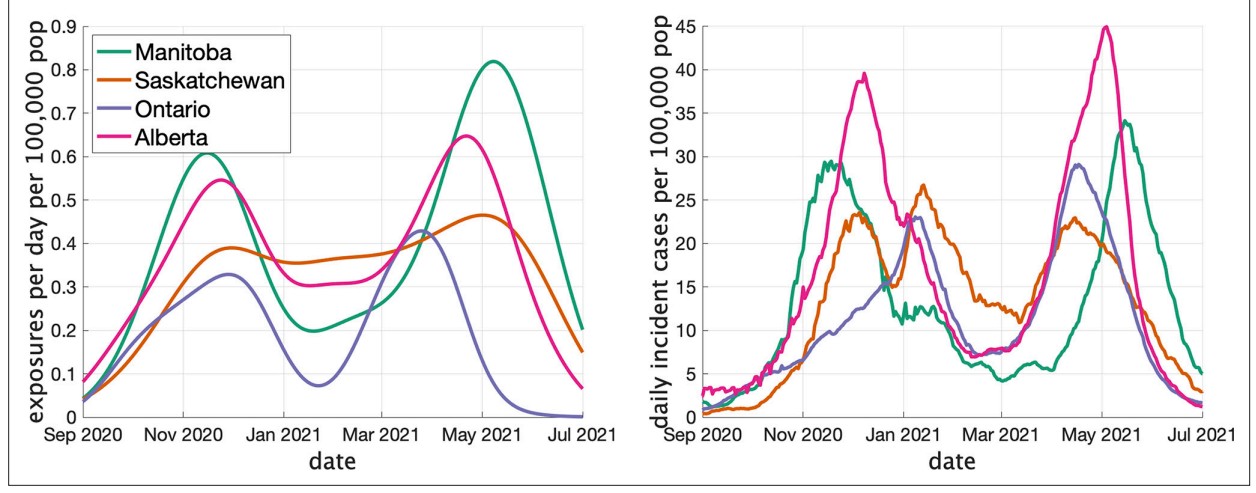

**Figure 3.** Two indicators of COVID prevalence over time in the four Canadian provinces. (Left) Estimates of the rate of new clusters (per 100,000 population) as a function of time in each province. (Right) Incident cases per day (per 100,000 population) in the same province over the corresponding time interval. Case counts are averaged over a 2-week window.

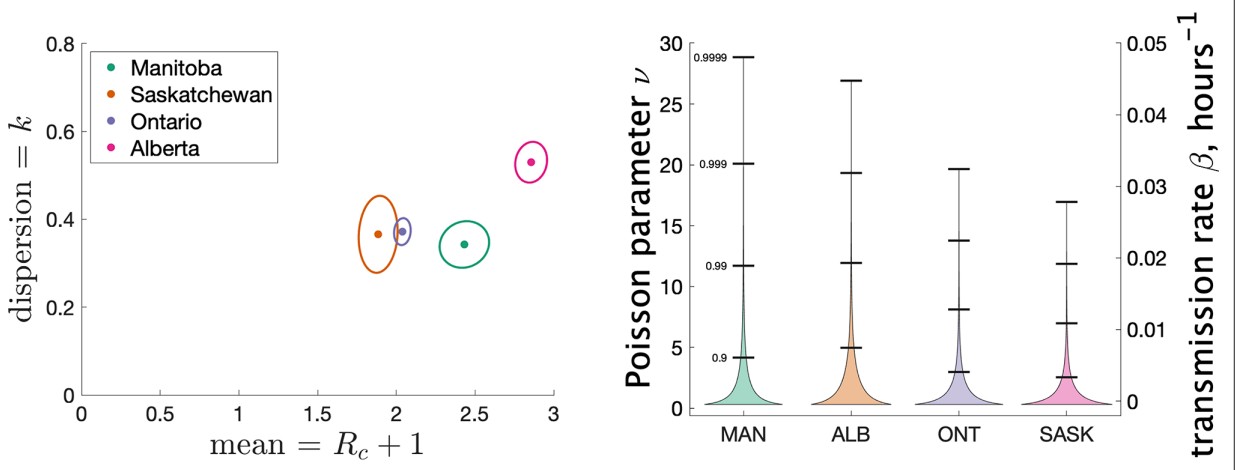

**Figure 4.** Results of our analysis for the four Canadian provinces. (Left) Estimates of mean and dispersion of cluster size for four Canadian provinces using the individual ascertainment model with ascertainment rate 0.75. Estimate of mean includes index case. The sample size for estimates for each province is the Number of Clusters as shown in *Table 1*. 95% confidence ellipses are shown, computed using the inverse Hessian method. (Right) Estimated distribution of $\nu$ (left axis) and instantaneous transmission rate $\beta$ (right axis) for different provinces.

many single-case clusters will not be detected at all, biasing the observed cluster size upwards again. We also consider an alternate model of ascertainment, where the chance of a cluster being reported at all depends on the size of the cluster, and vary the rate of ascertainment in that alternate model; see Appendix 1.

Another way to visualize the variability of transmission we have inferred from the data is to show the distribution of the Poisson parameter $\nu$, of which $R_c$ is just the mean. In our model $\nu$ is the index case-specific expected number of further cases in a cluster, and is a gamma-distributed random variable. *Figure 4* (right) shows the estimated distribution of $\nu$ for each jurisdiction, and *Table 2* shows some key properties of the distribution for each of the provinces.

As a way of interpreting dispersion values and what they mean for cluster size, we consider the fraction of all cases that occur in the largest 20% of all clusters. (If the distribution of cases follows the Pareto principle *Wikipedia contributors, 2021* then 80% of the cases will be in the top 20% largest clusters.) If we consider only secondary cases (not including the index case) we see from *Figure 5* (right) the fraction that are due to the 20% largest clusters for various values of mean cluster size and $k$. For example, for Alberta with a mean cluster size of 2.9 and a dispersion $k$ of 0.53, 69% of the secondary cases are in the top 20% of the clusters by size. For Saskatchewan, with a mean cluster size of 1.9 and $k = 0.37$, 82% of secondary cases are in the top 20% of clusters by size. When we include index cases, the fractions are correspondingly lower, as we see in *Figure 5* (right).

Our model does not consider the details of transmission at the individual level, and so does not make use of an instantaneous transmission rate per contact pair. However, by making some simple assumptions about SARS-CoV-2 transmission, we can infer a distribution of transmission rate $\beta$ from our estimate of the distribution of the parameter $\nu$. Recall that $\nu$ is a Gamma-distributed random

**Table 2.** Properties of the estimated distribution for the Poisson parameter $\nu$, the index case-specific expected number of further cases in a cluster.

The expected value of $\nu$ is $R_c$ and its distribution gives important information about overdispersion of clusters. In units of hours$^{-1}$.

| Province | Mean | Standard deviation | Median | 90th percentile | 99th percentile |
|---|---|---|---|---|---|
| Alberta | 1.86 | 2.55 | 8.9e-01 | 5.0 | 11.9 |
| Manitoba | 1.43 | 2.45 | 4.3e-01 | 4.1 | 11.7 |
| Saskatchewan | 0.88 | 1.46 | 2.9e-01 | 2.5 | 7.0 |
| Ontario | 1.04 | 1.70 | 3.5e-01 | 3.0 | 8.1 |

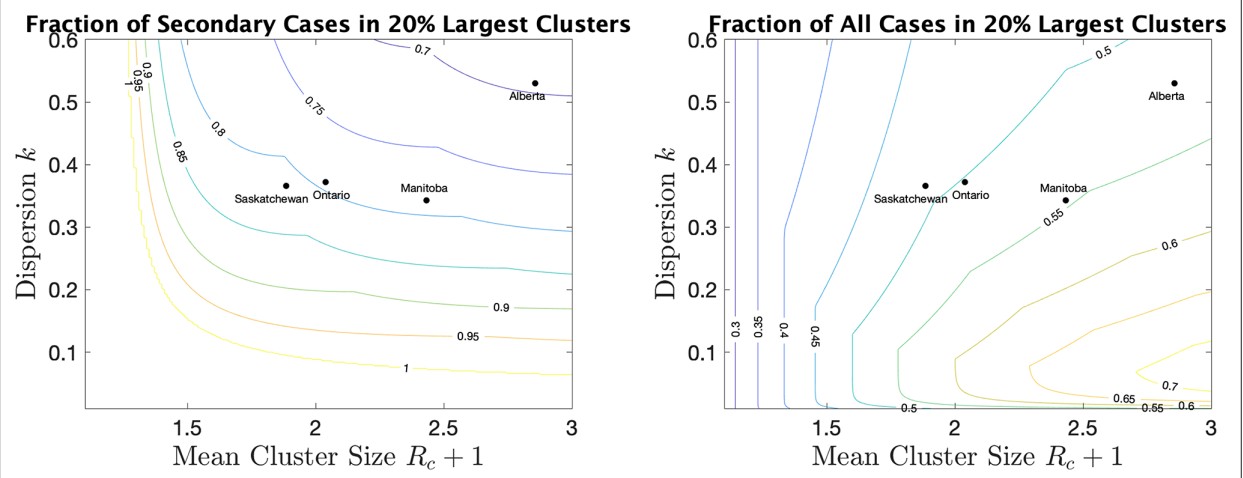

**Figure 5.** For a range of mean cluster size and dispersion $k$, the fraction of cases in the 20% largest clusters, counting only secondary cases (left), or all cases, index and secondary (right). Dots indicate the location of the four provinces in the plots.

variable that gives mean number of secondary cases. Another way to estimate mean cluster size is to use an individual contact model where when an infectious person is in contact with a susceptible person, the susceptible person is infected with rate $\beta$. In such a model we assume that infected individuals are in a classroom for 2 days before isolating (when they develop symptoms), and that the total contact time with their classmates is $T = 12$ hr. Assuming that all individuals are in the same class, the infected individual is in contact with $n = 25$ other susceptible students for that time period. Then the infected individual will on average infect $\beta nT$ other students. So we estimate $\beta = \nu/(nT)$. Since $\nu$ is Gamma-distributed, our estimate of $\beta$ is too. For estimating the distribution of $\beta$ we used a 4-day window for the definition of clusters, since this is more likely to include only people directly infected by the index case. *Figure 6* shows our estimated distribution of $\beta$ for the different Canadian provinces. *Table 3* shows some of the features of the estimated distribution for $\beta$.

One application of these estimates of the distribution of $\beta$ is that we can explore the consequences of different types of interventions in the classroom setting. In *Tupper et al., 2020* the authors consider a simple model of SARS-CoV-2 transmission among a group of contacts and investigate the quantity $R_{event}$, the average number of secondary infections due to the presence of a single infectious individual. $R_{event}$ is determined by $T$, the total length of time the infectious individual is with others; $n_{contact}$, the number of contacts at any point in time; $\tau$ the length of time the individual is with a fixed set of contacts; and $\beta$, the instantaneous transmission rate. The parameter $\tau$ can vary between some fraction of $T$ (e.g., $T/3$, if the index case divides their time equally between three sets of $n_{contact}$ contacts) or $T$ if the set of contacts is fixed. Interventions can be classified according to which of these parameters they modify: reducing transmission reduces $\beta$, social distancing reduces $n_{contact}$, and 'bubbling' (staying with the same small group rather than mingling) increases $\tau$ to $T$. If we use our distributions for $\beta$ with the model of *Tupper et al., 2020* we can estimate how the distribution of cluster sizes is changed with different interventions under different values of the parameters $R_c$ and $k$.

In *Figure 7* we show estimated size distributions of clusters under different interventions. Our baseline simulation settings intend to capture a pre-COVID high school classroom: $T = 12$ hr (2 days of exposure before the index case isolates), $\tau = 3$ hr (each student has four different classes that they attend for equal periods of time), $n_{class} = 25$, and $\beta$ is sampled from our estimated distribution for a given choice of $R_c$ and $k$. We consider three interventions: transmission reduction (e.g., by introducing masks) reduces $\beta$ by a factor of 2; social distancing cuts the size of a class in half; strict bubbling increases $\tau$ to $T$. For all values of $R_c$ and $k$ we consider, we simulate $10^7$ clusters to obtain a histogram of the number of secondary cases as well a mean and standard deviations, for the baseline conditions and for each of the three interventions, as shown in *Figure 7*. Means and standard deviations are accurate to the number of digits reported, and are shown with the corresponding histogram in the figure.

*Figure 7* (left) shows results for $R_c$ and $k$ close to that of Manitoba with a 4-day window for cluster definition ($R_c = 1.0$, $k = 0.4$). We see that both reducing transmission and social distancing are effective

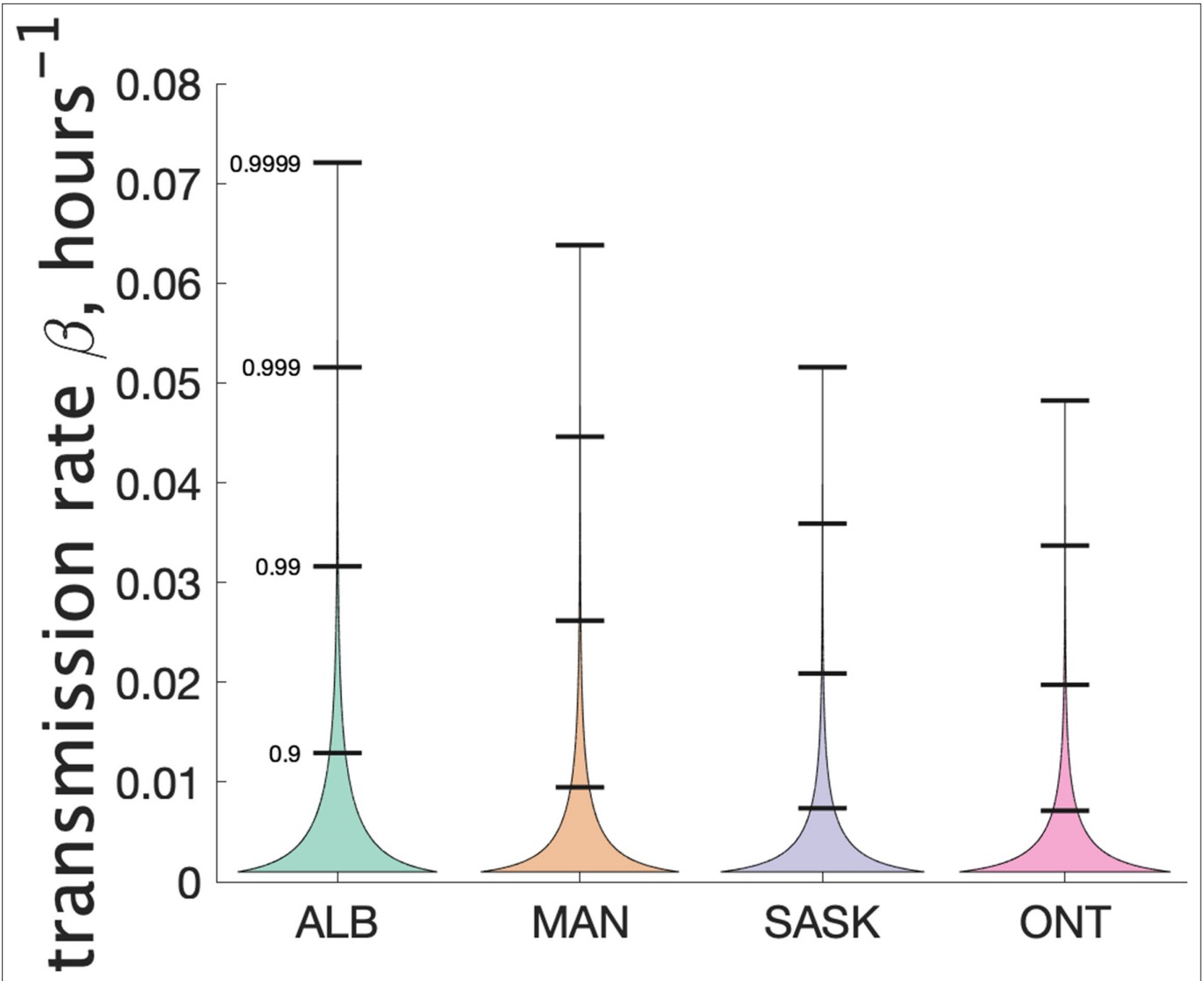

**Figure 6.** Estimated distribution of $\beta$ for different provinces.

in reducing the total number of cases, whereas bubbling does not contribute much to reduce cluster sizes. This is characteristic of what (*Tupper et al., 2020*) call the linear regime: the number of secondary infections depends linearly on the time the infectious person is present with others. *Figure 7* (right) shows the results in a hypothetical setting where $R_c$ is much larger ($R_c = 2.5$, $k = 0.4$), perhaps due to the existence of a more transmissible variant such as Omicron. Here, transmission reduction is less effective than in the linear regime, and strict bubbling more so; increasing $\beta$ has moved us closer to the so-called saturating regime, where transmission reduction is relatively less effective than bubbling.

**Table 3.** Properties of the estimated distribution for the instantaneous transmission rate $\beta$. In units of hours$^{-1}$.

| Jurisdiction | Mean | Standard deviation | Median | 90th percentile | 99th percentile |
|---|---|---|---|---|---|
| Alberta | 4.8e-03 | 6.7e-03 | 2.2e-03 | 1.3e-02 | 3.2e-02 |
| Manitoba | 3.3e-03 | 5.5e-03 | 1.1e-03 | 9.5e-03 | 2.6e-02 |
| Saskatchewan | 2.5e-03 | 4.4e-03 | 7.4e-04 | 7.4e-03 | 2.1e-02 |
| Ontario | 2.5e-03 | 4.1e-03 | 7.8e-04 | 7.1e-03 | 2.0e-02 |

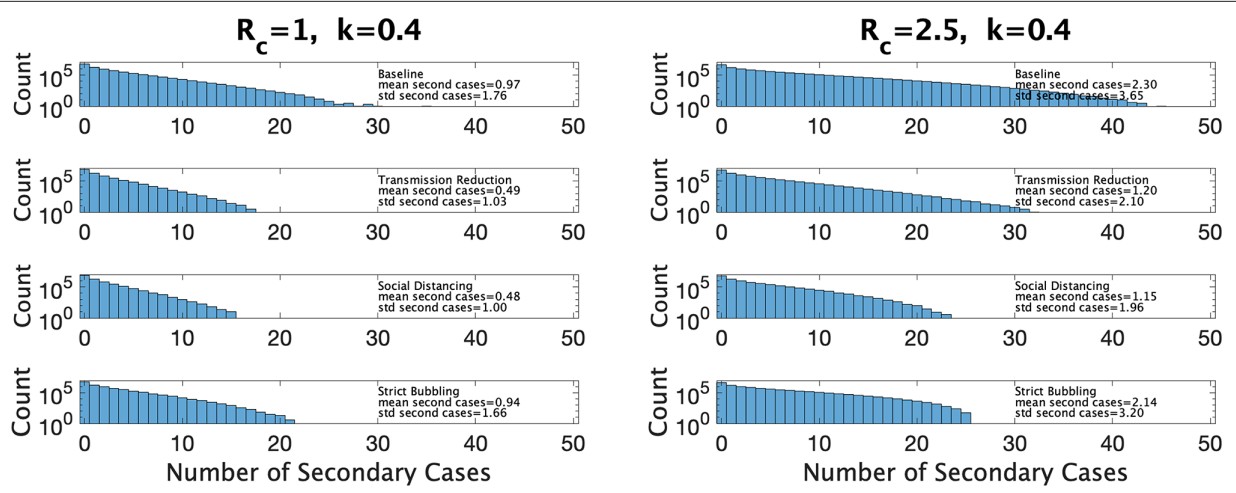

**Figure 7.** Distribution of the number of secondary infections under baseline conditions and under three interventions. Left: under parameter choice $R_c = 1$ and $k = 0.4$. Right: with $R_c = 2.5$.

## Discussion

We have used cluster size data to estimate the mean and dispersion in cluster sizes, accounting for imperfect case detection. We have found that in each of the provinces we consider, the majority of school transmission occurs in a small number of classrooms, with the top 20% of clusters containing between 70% and 80% of the secondary cases in school settings. We developed a method to estimate the transmission rate per contact per unit time, with reference to a simple model of classroom transmission. Having a direct estimate of the transmission rate allows us to compare the benefits of different control measures. We find that with parameters estimated from Canadian jurisdictions during the 2020–2021 school year, interventions that reduce transmission rates (such as masking) and reduce number of contacts at any one time (class size reduction), are more effective than strategies aimed at keeping sets of contacts consistent (such as bubbling).

Overdispersion in transmission of SARS-CoV-2 and other infectious diseases is well documented (e.g., *Woolhouse et al., 1997*) and is often described with reference to the 20/80 rule: that 20% of the infected individuals account for 80% of the transmission. Naturally, if the more infectious 20% can be identified, interventions targeting that portion of the population are likely to have a high impact. For SARS in 2003, *Lloyd-Smith et al., 2005*, estimated that 20% of the cases were responsible for almost 90% of the transmission. Estimates for SARS-CoV-2 also find considerable overdispersion, with the parameter $k$ between 0.1 (*Endo et al., 2020*) and 0.5 (*Laxminarayan et al., 2020*) (with $R_0 = 2.5$ this gives the top 20% of cases causing 69–96% of the transmissions; see *Sneppen et al., 2021*, for a survey). These estimates focus on the distribution of the number of people an infectious person infects directly during the whole course of infection (with mean $R_0$), which is of obvious epidemiological importance, but for which it is difficult to obtain high-quality data. When a case is identified, we are not always able to determine who they infected, and indirect methods must be used. We may miss cases, and others may be wrongly attributed to a given index case.

In our present study, we examined a different random quantity, the number of additional cases $Z$ infected, either directly or through intermediaries, by a given index case in a given setting. We denoted the mean of $Z$ by $R_c$. Including the index case means that the cluster size is $Z + 1$, with mean $R_c + 1$. Compared to estimates of $R_0$, $R_c$ does not count people infected at other sites, but it does include additional cases, because it includes both direct and indirect transmission. $Z$ and its mean $R_c$ are therefore more focused on the particular setting (in this case a school) than $R_0$ is. In general it will depend on the infectiousness of the index case, as well as how conducive the environment is to transmission, and what activities are undertaken there. Determining the distribution of $Z$, as we have done here, provides an alternative means of investigating transmission.

However, these two measures of transmissibility ($R_0$ and $R_c$, the mean of $Z$) may be close enough that it is instructive to compare our estimates for $Z$ with the traditional $R_0$, and our dispersion estimates

with dispersion estimates for the number of secondary infections. Our $R_c$ ranges from 0.9 in Saskatchewan to 1.9 in Alberta. These low values of $R_c$ are inconsistent with $R_0$ estimates (which range from 2 to 6; *Alimohamadi et al., 2020*), and indicate that in the pre-Delta time frame in these jurisdictions schools were unlikely to be a major contributor to SARS-CoV-2 spread. However, with increased transmissibility with new variants such as Omicron, this situation may have changed. The discrepancy is even greater when we consider clusters defined by the 4-day window, which are even smaller. Our estimates for $k$ range from 0.34 (in Manitoba) to 0.53 (in Alberta), corresponding closely to earlier estimates of dispersion.

Overdispersion has consequences for controlling transmission and for estimation. Estimating the average transmission rate from a small number of clusters will be difficult, and will result in a high variability. Most likely what will be observed in a small number of sampled clusters will be little to no onward transmission, which would lead to underestimates of the transmission rate. But if one or more larger clusters are included in a sample by chance, then this could lead to an overestimate of the transmission rate.

If we could identify the conditions under which the rare larger clusters occur (high-risk individuals, activities, and settings) we could achieve a disproportionately large effect on reducing transmission by applying new measures in these settings. There are myriad possible reasons for overdispersion of transmission for SARS-CoV-2, including variation in viral load (*Chen et al., 2020*), behaviour, and number of contacts. But a key factor in higher dispersion with SARS-CoV-2 in comparison to other pathogens such as influenza is aerosolization (*Goyal et al., 2021*), which allows the index case to infect others in the room even if they are not a close contact. Properties of the setting may be very important, with some settings (cramped, poor ventilation) being especially conducive to transmission. It would be good to identify classrooms or schools where there is a high risk of larger clusters. For example, if data were available on the occupancy, ventilation standards, mask use, classroom size, distancing behaviour, and other features of classrooms, we could investigate how this related to the cluster size. Rapid tests may be especially good for identifying the most infectious individuals, given that they are sensitive to viral loads (*Mina and Phillips, 2021*), but additional data collection is likely needed to quantify setting-level risks.

Two important changes have happened since the majority of the data here was collected. Firstly, in the jurisdictions studied, effective vaccines have been developed and deployed for those aged 5 and up (*The New York Times, 2020*). There are several ways in which this may effect cluster sizes in the school setting. To the extent that the general population (including adults) being vaccinated reduces incidence of COVID (*Wilder-Smith and Mulholland, 2021*; *Leshem and Wilder-Smith, 2021*; *Mallapaty, 2021*), there will be fewer introductions of SARS-CoV-2 into the classroom, and so fewer exposures will occur leading to fewer clusters. This effect may be dampened by relaxation of distancing and other measures that were keeping COVID-19 at bay and are no longer necessary in the context of vaccination. The distribution of cluster sizes when clusters do occur will also change: many students who might otherwise be infected will be protected by the vaccine, others who are vaccinated but infected (breakthrough infection) may have reduced symptoms and therefore may not be detected. We therefore expect the mean cluster size to be reduced by vaccination, in age ranges where vaccination has been deployed. It is unclear what the consequences will be for the dispersion.

Secondly, new, more transmissible variants of SARS-CoV-2 have emerged (*CDC, 2021a*), most notably the Alpha variant, the Delta variant, the Omicron variant, and most recently the BA.2 strain of the Omicron variant, each with a substantially higher transmissibility than its predecessors. A natural way to implement this change in our model is to multiply $R_c$ by an appropriate factor, boosting the size of clusters, without changing the dispersion parameter $k$. Data from the period in which Delta was the prevalent strain is not available, but schools in the Canada and the US saw resurgences in clusters in schools around school openings (*Cravey, 2021*; *Star staff wire services, 2021*; *CNN, 2021*).

Our data and model have some limitations. The data rely on crowdsourcing, and there is reason to believe that reporting is incomplete. Inequity may effect data collection, as wealthier districts are more likely to have the resources to identify and track transmission. In general, larger clusters may be more likely to be reported. In the modelling, we assumed a Poisson random variable for the cluster size, with an underlying gamma-distributed rate variable. This is a flexible model allowing for considerable overdispersion, but it is simple and does not explicitly handle complexities such as the differences between elementary and high schools. Our estimates of the transmission rate were derived

(where feasible) from a model with a fixed number of hours that the index case would be infectious in the classroom, and fixed class sizes. Accounting for variation in these would result in more variability in the estimates.

A major limitation of our analysis is how we assigned cases to clusters. Since the only data available was the number of cases reported on a given day at a school, we put cases in the same cluster if they occurred within 7 days of each other. The choice of 7 days was informed by the serial interval of COVID-19, but unavoidably, some cases will have been put in clusters that were not linked by transmission, whereas other that were linked were not put in the same cluster. Furthermore, we assumed that all clusters consisted of an index case and a number of other cases directly infected by the index case. In reality, there may be longer chains of transmission. Any of these assumptions may bias our estimates of the distribution of $\nu$ and $\beta$. Finally, our illustrative modelling of the impact of interventions was simple, and used simple assumptions for the impacts of masking, distancing, and cohorts (bubbling). Our estimates of the per-contact transmission rate per unit time could, however, be used in more sophisticated simulation modelling to compare interventions.

Despite these limitations, our approach has distinct advantages. We have developed estimates of the person-to-person transmission rate derived directly from data. The data we use (cluster sizes) are relatively easy to access. This approach does not require individual-level data or contact tracing information, which are often not available; individuals may be identifiable and data are held within public health institutions. However, we note that if it were available, contact tracing data would be an excellent gold standard against which to check our assumptions about cluster identification. Our estimation approach, together with cluster size data, offers a high-resolution view of transmission: we can estimate the distribution of cluster sizes in specific settings, accounting for reporting and overdispersion, and in some contexts we can estimate the transmission rate, all without requiring either individual-level data or assumptions on transmission parameters such as the serial interval (see, in contrast, *Cori et al., 2013*; *Wallinga and Teunis, 2004*, which require serial interval estimates). The results offer context-specific tools to simulate interventions in particular settings (here, schools). The methods are readily generalizable to other structured settings, such as workplace outbreaks where workplaces are similar in size and structure. Our results also suggest the need for data collection activities that can relate cluster sizes to setting variables such as occupancy, density, ventilation, activity, and distancing behaviour. Ultimately this would provide the data needed to design interventions that best reduce school and/or workplace transmission.

## Acknowledgements

We thank Alisha Morris and the other volunteers at the National Education Association for providing the US data that was used in this study. PT and CC were supported by Natural Science and Engineering Research Council (Canada) Discovery Grants (RGPIN-2019-06911 and RGPIN-2019-06624). CC receives funding from the Federal Government of Canada's Canada 150 Research Chair program.

## Additional information

### Group author details

**COVID Schools Canada**
**Urooj Khan**: The Hospital for Sick Children, Toronto, Canada; **Ololade Ogunsuyi**: Georgetown University, Washington DC, United States; **Hinda Hussein**: McMaster University, Hamilton, Canada; **Andreea Manea**: York University, Toronto, Canada; **Joshua Atienza**: University of Toronto, Toronto, Canada; **Rafa Abbas**: Cumming School of Medicine, University of Calgary, Calgary, Canada; **Syeda J Hasan**: Schulich School of Medicine and Dentistry, University of Western Ontario, Waterloo, Canada; **Ahmed Aldarraji**: Michael G. DeGroote School of Medicine, McMaster University, Hamilton, Canada; **Anjalee Benedict**: York University, Toronto, Canada; **Ahmed Al-Jaishi**: Lawson Health Research Institute, London, Canada

## Competing interests
Caroline Colijn: Reviewing editor, eLife. The other authors declare that no competing interests exist.

## Funding

| Funder | Grant reference number | Author |
| --- | --- | --- |
| Natural Sciences and Engineering Research Council of Canada | RGPIN-2019-06911 | Paul Tupper |
| Natural Sciences and Engineering Research Council of Canada | RGPIN-2019-06624 | Caroline Colijn |
| Government of Canada | Canada 150 Research Chairs Program | Caroline Colijn |

The funders had no role in study design, data collection and interpretation, or the decision to submit the work for publication.

## Author contributions
Paul Tupper, Conceptualization, Data curation, Software, Formal analysis, Investigation, Visualization, Methodology, Writing – original draft, Writing – review and editing; Shraddha Pai, Resources, Data curation; COVID Schools Canada, Data curation, Investigation; Caroline Colijn, Conceptualization, Formal analysis, Investigation, Methodology, Writing – original draft

## Author ORCIDs
Paul Tupper  http://orcid.org/0000-0002-4340-4481
Shraddha Pai  http://orcid.org/0000-0002-1048-581X
Caroline Colijn  http://orcid.org/0000-0001-6097-6708
Urooj Khan  http://orcid.org/0000-0001-5151-9943
Ololade Ogunsuyi  http://orcid.org/0000-0002-1718-9829
Hinda Hussein  http://orcid.org/0000-0002-0531-8128
Andreea Manea  http://orcid.org/0000-0002-6844-3313
Joshua Atienza  http://orcid.org/0000-0001-7452-7268
Rafa Abbas  http://orcid.org/0000-0002-2011-6815
Syeda J Hasan  http://orcid.org/0000-0001-5930-3282
Ahmed Aldarraji  http://orcid.org/0000-0002-5985-1702
Anjalee Benedict  http://orcid.org/0000-0002-4957-7543
Ahmed Al-Jaishi  http://orcid.org/0000-0003-0376-2214

## Decision letter and Author response
Decision letter https://doi.org/10.7554/eLife.76174.sa1
Author response https://doi.org/10.7554/eLife.76174.sa2

# Additional files

## Supplementary files
• Transparent reporting form

## Data availability
Code and data have been deposited in GitHub (https://github.com/PaulFredTupper/covid-19-clusters-in-schools, copy archived at swh:1:rev:77cc5d7f42cde3c4eb71500b52a9797f6762712e) and Zenodo (https://doi.org/10.5281/zenodo.7117270).

The following dataset was generated:

| Author(s) | Year | Dataset title | Dataset URL | Database and Identifier |
|---|---|---|---|---|
| Tupper P, Colijn C, Pai S, Khan U, Ogunsuyi O, Hussein H, Manea A, Atienza J, Abbas R, Hasan SJ, Aldarraji A, Benedict A, Al-Jaishi A, SOS Alberta, Drouin O, BC School Covid Tracker, Morris A, National Educational Association | 2021 | Crowdsourced COVID-19 cases and outbreaks in US and Canadian Schools in 2020-21 | https://doi.org/10.5281/zenodo.7117270 | Zenodo, 10.5281/zenodo.7117270 |

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

## Appendix 1

### Model for cluster size

We consider two models for ascertainment (whether a case is actually detected), though we only consider the first in the main text.

In the first ascertainment model (individual ascertainment) each of the infected individuals is detected with a probability $q_1$. So $X$, the total number of infected individuals is binomial $(n, p)$ with parameters $n = Z + 1$ and $p = q_1$. If by chance none of the individuals are observed, we do not observe the cluster. This is meant to model a situation where cases are detected independently of each other, and one detected case does not lead to further tests or screening.

In the second ascertainment model (cluster ascertainment), at first each case is identified with probability $q_2$, but then if any of the students are identified they are all subsequently identified. This is intended to capture a situation where a single detected case triggers testing for the whole class. Again, if no cases are detected we do not observe the cluster. This is equivalent to saying that clusters of size $m$ are detected in their entirety with probability $1 - (1 - q_2)^m$.

The number $Z$ of new cases given the presence of one infectious case is a Poisson-distributed random variable with a rate $\nu$ that is itself a Gamma-distributed random variable with a shape $k$ and scale $\theta$. This means $Z$ has a negative binomial distribution $\mathrm{NB}(r, p)$, where $r = k$ and $p = \theta/(1 + \theta)$. Letting $\Theta = (k, \theta)$, the pmf of $Z$ is

$$V_\Theta(j) = P(Z = j) = \tfrac{\Gamma(j+r)}{j!\Gamma(r)}(1 - p)^r p^j.$$

Under the individual ascertainment model with ascertainment probability $q_1$, $X$, the number of observed cases, is binomial $(n, p)$ with $n = Z + 1$ and $p = q_1$. So, the probability that $i$ individuals are observed is

$$W_{\Theta,q_1}(i) = \sum_{j=0}^{\infty} \binom{j+1}{i} V_\Theta(j) q_1^i (1 - q_1)^{j+1-i}$$

for $i = 0, 1, \ldots$. Since we do not observe clusters with no observed cases the probability of observing a cluster of size $i$ is actually $P(X = i) = C_{\Theta,q_1}^{-1} W_{\Theta,q_1}(i)$ for $i = 1, 2, \ldots$, where $C_{\Theta,q_1} = \sum_{i=1}^{\infty} W_{\Theta,q_1}$.

If the observed cluster sizes are $X_i$, $i = 1, \ldots, n$, the log-likelihood function for $\Theta = (k, \theta)$ under the individual ascertainment model is then

$$\sum_{i=1}^{n} \log \left[ C_{\Theta,q_1}^{-1} W_{\Theta,q_1}(X_i) \right]$$

Under the cluster ascertainment model, the cluster is observed or not with probability $1 - (1 - q_2)^{Y+1}$. So the probability of observing $X = j$ in a cluster is
$U_{\Theta,q_2}(i) = [1 - (1 - q_2)^i] V_\Theta(i)$ for $i = 0, 1, \ldots$ but then since we cannot observe clusters of size 0, an observed cluster has size $i$ with probability
$P(X = i) = D_{\Theta,q_2}^{-1} U_{\Theta,q_2}(i)$ for $i = 1, 2, \ldots$, where $D_{\Theta,q_1} = \sum_{i=1}^{\infty} U_{\Theta,q_2}(i)$.

If the observed cluster sizes are $X_i$, $i = 1, \ldots, n$, the log-likelihood function for $\Theta = (k, \theta)$ under the cluster ascertainment model is then

$$\sum_{i=1}^{n} \log \left[ D_{\Theta,q_2}^{-1} U_{\Theta,q_2}(X_i) \right]$$

Under both ascertainment models, we then go from our estimates of $k$ and $\theta$ to estimates of $R_c$ via the formula

$$R_c = \tfrac{pr}{1-p} = k\theta.$$

We use the Delta method to obtain confidence intervals for $R_c$ from confidence intervals on $k$ and $\theta$.

### Analysis of US data

The US data was gathered from the National Educational Association website (*Canadian Doctors, Professionals, & Citizens for Masks, 2021*) (originally started by Alisha Morris, an educator at a Kansas high school) which collected data from news media and from reports submitted by volunteers

(*Walker, 2021*). We selected the eight states with the most data available, and covering the period between August and November 2020. For the US data we used confirmed student cases listed on a particular date for the cluster size, excluding teachers and staff. We did not collect cases reported on different days at the same school in the same cluster as we did with the Canadian data.

In *Appendix 1—table 1* we show some statistics associated with the data for each state. In the top we show the number of clusters, the number of schools appearing, the number of schools with more than one reported cluster, and the fraction of schools with multiple clusters. In the bottom we show the fraction of clusters that have only one observed case, and the average number of observed cases in the clusters, the maximum observed cluster size, the index of dispersion (variance divided by mean) of cluster size, and index of dispersion of the number of cases in a cluster subtracting one for the presumed index case. Comparing with *Table 1*, we can see several striking differences between the US and Canadian data. There are substantially more clusters reported in Canada than in the US, despite the US states having greater population on average. This may partly be explained by the Canadian data being collected over a longer period than the US data, but this is likely not the full explanation: in *Appendix 1—figure 2* we show the rate (in clusters per day) that cases appear in the dataset over time. We can compare with *Figure 3* (left) that shows the same thing for the Canadian data. Even at times when both US and Canadian datasets record clusters, Canadian rates are higher than US rates by an order of magnitude, despite incidence rates being similar in US states versus Canadian provinces (*Appendix 1—figure 2* (right) versus *Figure 3* (right)). This suggests that the method used for gathering cluster reports in different jurisdictions varied substantially between the two datasets, especially when we look at daily incident cases in each. Furthermore, the majority of schools in the US datasets only report one cluster., whereas the opposite is true of the Canadian data.

There are also substantial differences in statistics of cluster sizes. Mean observed cluster sizes were without exception larger in the US states than Canadian provinces, and Canadian provinces tended to have a higher fraction of clusters with only one case. Given the incomplete nature of the US data, we cannot determine whether these differences are due to real differences in transmission in the jurisdictions, or because smaller clusters were less likely to be reported in the US states.

**Appendix 1—table 1.** Cluster statistics for each state in the US data.
(Top) For each of the eight US states: number of clusters in the data, number of schools reported, number of schools with multiple clusters, fraction of schools with multiple clusters. (Bottom) Fraction of clusters with one case, mean observed cluster size, maximum observed cluster size, and index of dispersion (variance of number of cases divided by mean number of cases) with and without subtracting one for the index case.

| State | Number of Clusters | Number of Schools | Schools with Multiple clusters | Fraction of Multiple schools clusters |
|---|---|---|---|---|
| Texas | 369 | 326 | 30 | 0.09 |
| Florida | 147 | 134 | 10 | 0.07 |
| Ohio | 122 | 95 | 12 | 0.13 |
| Pennsylvania | 322 | 247 | 46 | 0.19 |
| Wisconsin | 130 | 118 | 11 | 0.09 |
| Georgia | 68 | 53 | 10 | 0.19 |
| Indiana | 105 | 84 | 13 | 0.15 |
| Illinois | 81 | 76 | 5 | 0.07 |

| State | Fraction with One case | Mean observed Cluster size | Max observed Cluster size | Index of Dispersion (IoD) | IoD without Index case |
|---|---|---|---|---|---|
| Texas | 0.69 | 1.73 | 21 | 2.29 | 5.40 |
| Florida | 0.69 | 5.31 | 90 | 27.28 | 33.62 |
| Ohio | 0.69 | 2.11 | 21 | 4.30 | 8.18 |

*Continued on next page*

*Continued*

| State | Fraction with One case | Mean observed Cluster size | Max observed Cluster size | Index of Dispersion (IoD) | IoD without Index case |
|---|---|---|---|---|---|
| Pennsylvania | 0.73 | 1.75 | 28 | 3.84 | 8.95 |
| Wisconsin | 0.46 | 4.02 | 31 | 6.28 | 8.36 |
| Georgia | 0.46 | 4.76 | 38 | 10.09 | 12.77 |
| Indiana | 0.81 | 2.19 | 36 | 9.00 | 16.56 |
| Illinois | 0.52 | 3.47 | 36 | 11.07 | 15.55 |

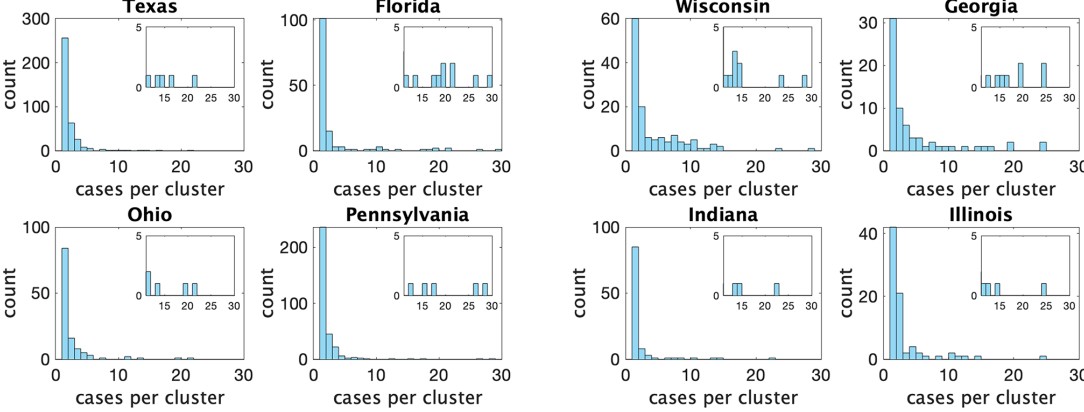

**Appendix 1—figure 1.** Histograms of observed cluster sizes in eight US states. We only show clusters of size 30 or fewer. Each dot represents a single cluster of size 8 or larger, and indicates the presence of (more rare) larger clusters.

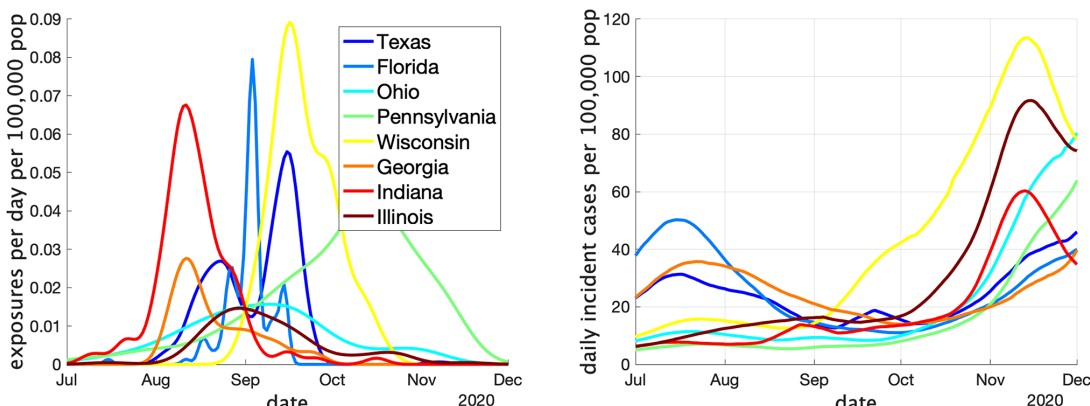

**Appendix 1—figure 2.** Two indicators of COVID prevalence over time in the eight US states. (Left) Estimates of the rate of new clusters being reported (per 100,000 population) as a function of time in each province. (Right) Incident cases per day (per 100,000 population) in the same province over the corresponding time interval. Case counts are averaged over a 2-week window.

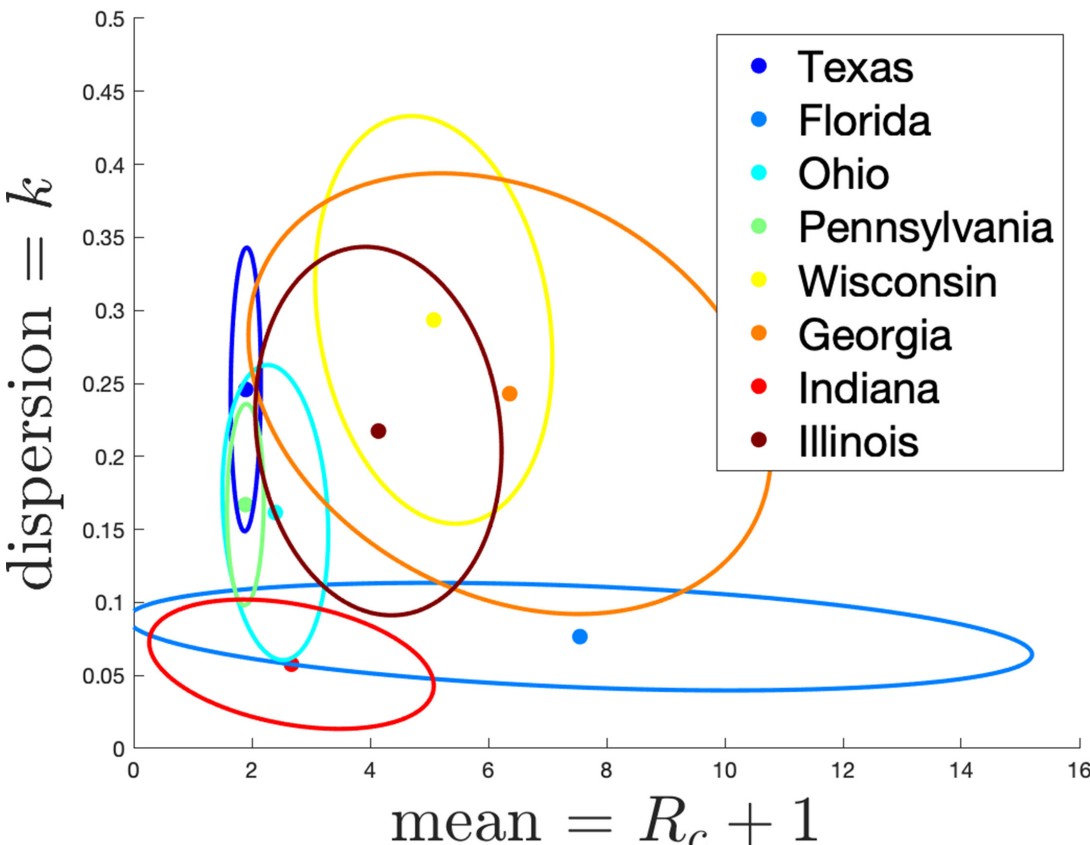

**Appendix 1—figure 3.** Estimates of mean and dispersion of cluster size for eight American states using the individual ascertainment model with ascertainment rate 0.75. Estimate of mean includes index case. The sample size for estimates for each state is the Number of Clusters as shown in *Appendix 1—table 1*. 95% confidence ellipses are shown, computed using the inverse Hessian method.

*Appendix 1—figure 3* shows the estimated mean cluster size (= $R_c + 1$) and dispersion $k$ same for the eight US states. In the US data, mean cluster size was estimated to range from about two in Texas to almost eight in Florida. Dispersions ranged from 0.05 to 0.3, showing considerable overdispersion compared to the Poisson distribution. However, given that we are very probably substantially undersampling clusters in the US data, and the clusters that we are observing are likely larger ones, these estimates of mean cluster size are biased upwards in a way we are not able to control for.

## Varying the rate and model of ascertainment

In the main text we estimated parameters with the assumption of the individual ascertainment model with an ascertainment probability of 0.75. Here, we investigate how our main parameters $R_c + 1$ (expected cluster size) and $k$ (dispersion) vary with this ascertainment probability. We also consider the alternate ascertainment model discussed in the previous section 'Model for cluster size'.

*Appendix 1—figure 4* shows the parameter estimates for the two models. The left plots show results for the individual ascertainment model where we set $q_2 = 1$ and vary $q_1$ from 0.2 to 1. The right plots show results for the group ascertainment model with $q_1 = 1$ and $q_2$ varying from 0.2 to 1. We see that the parameters do vary with the model and the ascertainment fraction, but relative magnitudes of the parameters in different jurisdictions do not change.

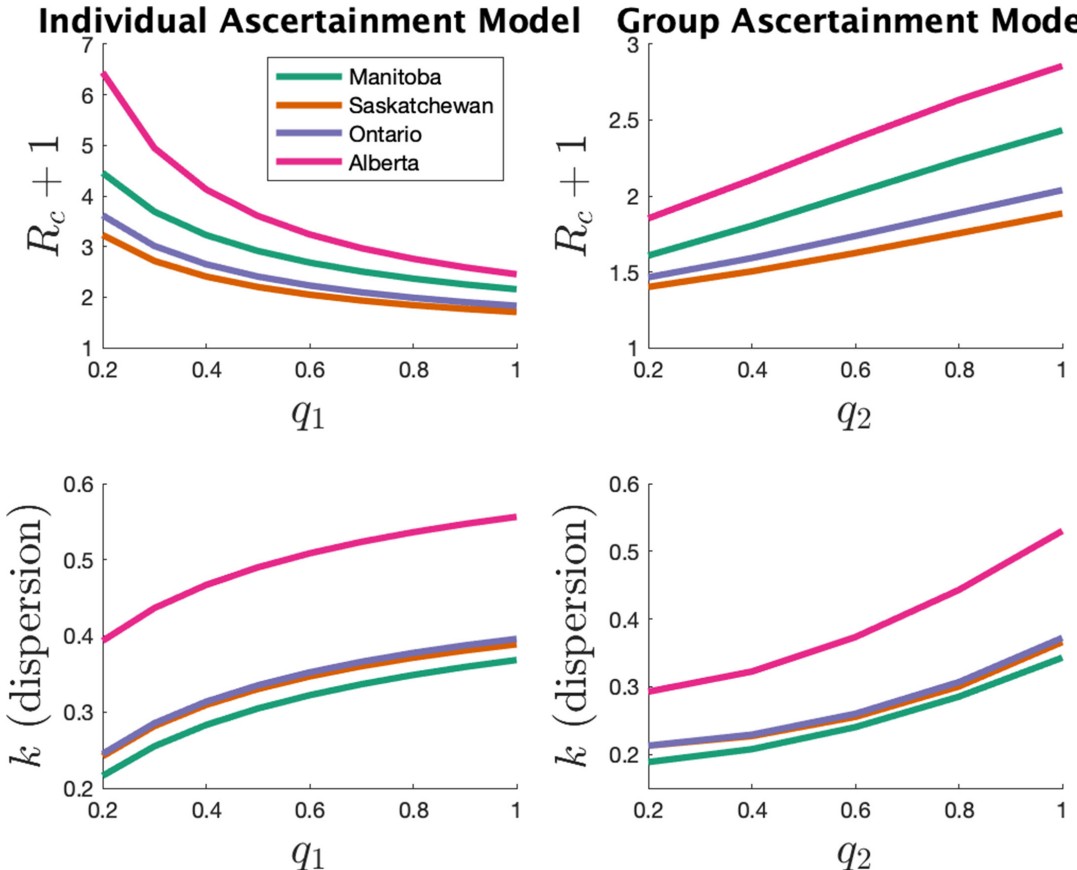

**Appendix 1—figure 4.** Estimates of mean and dispersion of cluster size for eight US states (left) and four Canadians provinces (right) using the individual ascertainment model (left) and the group ascertainment model (right) with varying ascertainment rate. Estimates of mean includes index case.

## Varying the window for assigning cases to a cluster

In the main text, we reported results when clusters were defined by assigning cases to the same cluster if they were reported within 7 days of each other, or if they could be linked by a chain of such cases. We investigate here how changing the window for defining clusters affects our results. In *Figure 5* we show how our estimates for dispersion ($k$) and mean cluster size ($R_c + 1$) vary when the window width is set to either 4, 7, or 10 days. Dispersion does not change much with changing the window, but as expected longer windows lead to larger clusters. However, the change in average cluster size from 7 to 10 days is modest, as we see in *Appendix 1—table 2*.

**Appendix 1—table 2.** Estimates of dispersion $k$ and mean cluster size $R_c + 1$ for the four provinces for three choices of the cluster definition window: 4 days, 7 days (the choice in the main text), and 10 days.

| Province | Dispersion $k$ | | | Mean cluster size $R_c + 1$ | | |
|---|---|---|---|---|---|---|
| | 4 days | 7 days | 10 days | 4 days | 7 days | 10 days |
| Manitoba | 0.36 | 0.34 | 0.35 | 1.99 | 2.43 | 2.71 |
| Saskatchewan | 0.34 | 0.37 | 0.36 | 1.76 | 1.88 | 1.98 |
| Ontario | 0.36 | 0.37 | 0.40 | 1.74 | 2.04 | 2.24 |
| Alberta | 0.50 | 0.53 | 0.54 | 2.43 | 2.86 | 3.16 |

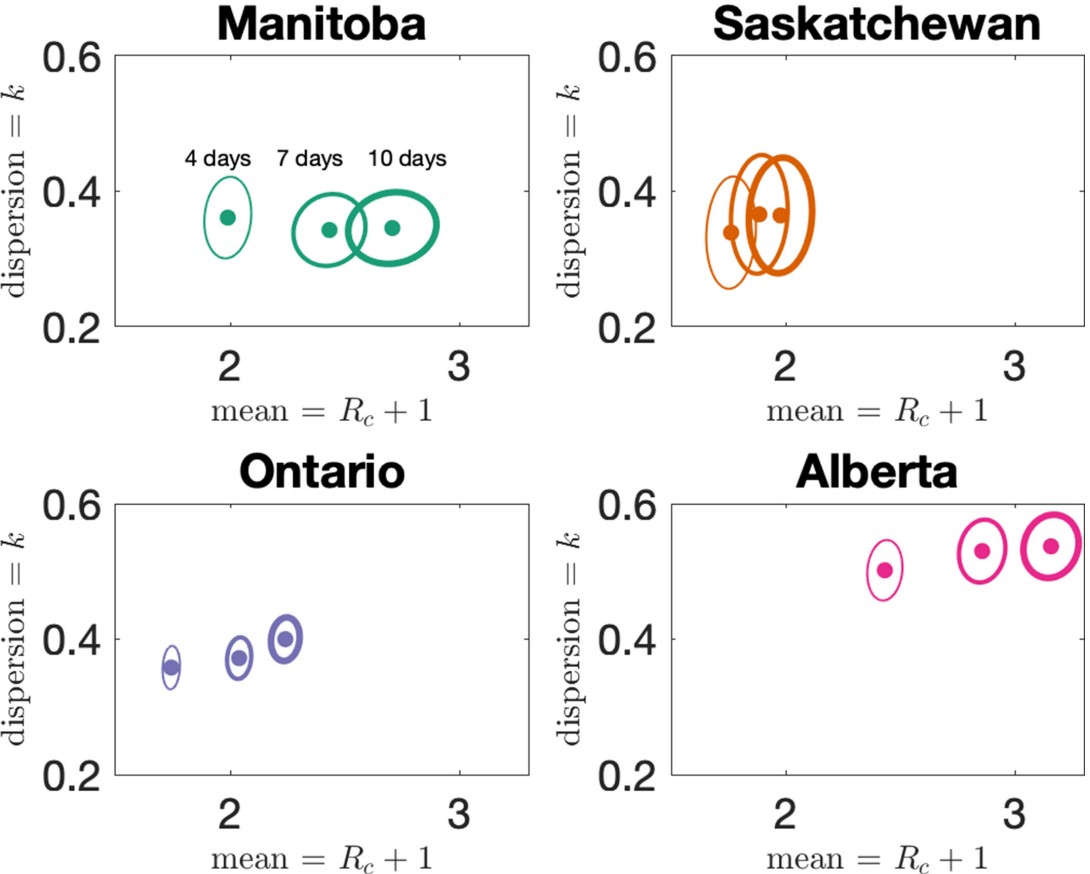

**Appendix 1—figure 5.** Estimates of mean and dispersion of cluster size for the four provinces with four different choices of the window for defining clusters. Choices are 4, 7, and 10 days, with thicker lines on the error ellipse indicating more days. The sample size for estimates for each province is the Number of Clusters as shown in *Table 1* of the main text. 95% confidence ellipses are shown, computed using the inverse Hessian method.

