## [Editor Report]

This paper provides an important novel methodology to understand the mode of spread of SARS-CoV-2 in schools given sparse data.

---

## [Decision Letter]

**Decision letter after peer review:**

Thank you for submitting your article "COVID-19 clusters in schools: frequency, size, and transmission rates from crowdsourced exposure reports" for consideration by *eLife*. Your article has been reviewed by 3 peer reviewers, including Joshua T Schiffer as Reviewing Editor and Reviewer #1, and the evaluation has been overseen by a Senior Editor.

The reviewers have discussed their reviews with one another, and the Reviewing Editor has drafted this to help you prepare a revised submission. Please submit a revised version that addresses these concerns directly. Although we expect that you will address these comments in your response letter, we also need to see the corresponding revision clearly marked in the text of the manuscript. Some of the reviewers' comments may seem to be simple queries or challenges that do not prompt revisions to the text. Please keep in mind, however, that readers may have the same perspective as the reviewers. Therefore, it is essential that you attempt to amend or expand the text to clarify the narrative accordingly.

Essential revisions:

1) Please reconstruct how the comparison between the US and Canadian data is done in the paper. All 3 reviewers agreed that the US data likely is hampered by overrepresentation of large super spreader events, leading to biased and inaccurate projections of the over-dispersion distribution. Please weight the 3 reviewers' suggestion for clarifying the presentation of this data.

2) Please either remove the time to detection analysis or incorporate a method to account for the inherent high degree of stochasticity of this outcome.

3) Please define cluster more clearly.

4) Please provide a more complete discussion about the true value of q and how this might impact model conclusions.

*Reviewer #1 (Recommendations for the authors):*

– Figure 1: x-axis needs labels and ticks to denote quantity.

– Please remove "not all the news is good for children": this language is too casual.

– All figures: increases font size substantially please.

– Figure 2: Please add an actual correlation plot with rho coefficients and p-values to formalize what appears to be true by eye. This will be so helpful to demonstrate that the US data is insufficient for the model used in the paper (Pardon the slang but… garbage in, garbage out). The authors can then emphasize the point that appropriately thorough data is necessary to model crowd sourced data of this nature.

– Table 1: given overdispersion, the median, interquartile range and full range of cluster size would be more informative than just the mean. Please add percentages to schools with multiple clusters column.

– Keeping in mind that this is a generalist journal., it would be helpful to show theoretical distributions with varying values of k as a supplemental figure. This would provide valuable context for Figure 3 and how to interpret k (in addition to Figure 4 which is great).

– The assumption that q=0.75 should be cited, or better yet, acknowledged up front as uncertain. The sensitivity analysis with varying values of q should be emphasized more when q is introduced.

– Figure 3: would it be possible to show the median and mean on the x-axis?

– Figure 4: why not include all provinces in Canada?

– Figure 5 is a bit difficult to interpret except for the most statistical minded among us, and could be a supplement (both the v and β parameters) or not included at all.

– For all figures comparing US and Canada data, please normalize the y-axes so comparisons can be made by eye more easily.

– When T < τ, how is”mingling” implemented? Does the number of contacts switch with movement from “classroom to classroom”. This needs greater explanation.

– Figure 7 is interesting and one of the coolest parts of the paper but lacks sufficient detail. For reduced β, increased social distancing and bubbling, the distribution is impacted to various degrees but is the total number of cases lower? Is the variance in the distributions lower? Are these differences statistically significant?

– Figure 7: The selection of R=0.5 and R=5 seems perhaps too extreme. Most persisting variants have been associated with R~1 (0.8-1.2) whereas omicron was 3-5 fold higher, albeit probably lower overall in classrooms. I would suggest at least testing an intermediate set of R values. Maybe 1 and 3?

– Please mention the omicron variant in the discussion.

– The authors make the excellent point that the crowd sourcing method is cheaper and easier than contact tracing or phylogenetic approaches but should also mention that contact tracing would be a wonderful gold-standard to validate the accuracy of crowd sourcing in terms of identifying cluster size.

– Figure 8: this would be easier to interpret of the panels were square rather than rectangular.

*Reviewer #2 (Recommendations for the authors):*

The major concern with the presented analysis is the use of sparse and potentially biased data in the analysis. The authors use compiled reports of case cluster sizes in schools across a number of jurisdictions, but have no methodological means to correct for issues clearly present in the underlying data. For example, most schools in the US only report a single cluster in the dataset, which likely biases the US dataset in a number of significant ways. This is contrasted with Canadian data where almost all schools report multiple clusters. The figures showing incident cases in schools and the community recapitulates this point with the US data showing no clear relationship between schools and the community compared with the Canadian data that shows a clear relationship. In either the case where school clusters contribute to transmission in the community or clusters are simply a byproduct of community transmission, we would expect to see correlation between the incident cases. While the authors mention these limitations to their study, I believe these data issues may present an existential issue with the current analysis as it makes it nearly impossible to interpret the results (even though I admit that the authors are fairly careful in their own framing). For example, it will be natural to wonder why Florida's Rc is nearly 8 while Texas's is around 2, and I believe there is very limited to say about the situation without controlling for underlying issues in the data between the states. I would suggest that the authors consider major methodological changes that might address the potential biases in the analysis. Perhaps limiting the analysis solely to Canadian data could also solve this issue, though there are likely biases in the dataset to consider as well.

The paper focuses on describing characteristics of clusters of cases in schools, and the authors give some details on the number of clusters in their datasets, however we never get a proper definition of what is considered a cluster here. According to the text on pages 11 and 12 it seems that an assumption is that all secondary cases in a cluster are infected by the same seed case (directly or indirectly). However, the interpretation of the results generally consider it to be through the direct route, even though public health policies would diverge significantly. For example a cluster that was infected outside of the school environment but were detected in school (imagine a group of close friends) would be counted in the analysis for Rc which ultimately filters to an understanding of how mitigation measures in school could control outbreaks. Alternatively, a cluster of multiple generations sputtering along might be given a large Rc when the actual R_eff for transmission might be below or close to 1.

The cluster definition is extra important, because it also impacts the estimate of the transmission rate. The transmission rate estimation assumes all transmission is happening within schools and in a single generation. In cases where there is actually a chain of transmission, or when transmission is happening outside of the school setting the formula used to calculate β does not hold. The authors partially acknowledge this by restricting the analysis to Canadian reports, but I think it should be more explicitly discussed in the manuscript either through an attempt to handle the complexity mathematically or with a larger discussion of the interpretation of the results.

Finally, the authors use an ascertainment q=0.75, which seems high, at least for the United States reports. For the US, the CDC estimates 1 in 4 infections were reported for a period covering the authors dataset, and detection is even less likely in children who are more likely to be asymptomatic (e.g. Davies et al.). The authors mention they used an alternate model of ascertainment and running a sensitivity analysis on q, where results for q=0.5 are potentially different from the main text results. Given the very possibility that the actual value of q is in that range, I would urge that the authors revisit their main results taking this into account.

*Reviewer #3 (Recommendations for the authors):*

Probably the most important observation here is that Canadian school cluster sizes and k parameters seem more consistent with a public health system that is accurately identifying (even small) school outbreaks, whereas the absence of single case outbreaks and the large fraction of very large outbreaks in the US data suggests a system which is calibrated to only identify (more obvious) superspreader events. Canada's per capita excess death rate during the pandemic has been notably lower (around 1/3) that seen in the US. Given that a relatively small contribution to mortality has come from pediatric cases, can the authors spell out a bit more explicitly how better surveillance may have blunted the pandemic's impact in Canada relative to the US?

Thank you for making the relevant data (https://github.com/PaulFredTupper/covid-19-clusters-in-schools/tree/main/schools_cluster_paper/datasets) and code accessible.

[Editors’ note: further revisions were suggested prior to acceptance, as described below.]

Thank you for resubmitting your work entitled "COVID-19 cluster size and transmission rates in schools from crowdsourced case reports" for further consideration by *eLife*. Your revised article has been evaluated by a Senior Editor and a Reviewing Editor.

The manuscript has been improved and is very close to satisfactory for acceptance but there are some remaining issues that need to be addressed, as outlined below:

1. Please make the abstract more balanced between methodologic gains and scientific conclusions.

2. Apply techniques to account for multiple generations of infection (or at least rerun the model assuming a smaller 4-5 day window to define clusters in line with the known serial interval) to ensure that this does not substantially impact the model's semi-quantitative conclusions.

*Reviewer #1 (Recommendations for the authors):*

Thank you for the extensive and extremely helpful changes to the article. It is well written, comprehensive and covers an important topic.

*Reviewer #2 (Recommendations for the authors):*

I greatly appreciate the authors revisions in response to the reviewer comments. In my previous review, I had a comment about the definition of the cluster and multiple generations of transmission. Now that the authors have provided that definition I have one major comment remaining.

The authors responded to my comment with: "These are indeed limitations of our analysis. We feel that we already addressed some of these limitations in the second-to-last paragraph of the discussion, but we have now added the following text as well: 'Furthermore, we assumed that all clusters consisted of an index case and a number of other cases directly infected by the index case. In reality, there may be longer chains of transmission. Any of these assumptions may bias our estimates of the distribution of ν and β.'"

Without strong evidence otherwise, I think the authors should address this concern within the analysis rather than simply a discussion. The definition of cluster now explicitly includes multiple transmission generations (through linking cases in successive weeks with one another). How many clusters have "multiple generations"? How does cluster size compare between single and multiple generation clusters? I imagine the multiple generation clusters are the largest ones, but it may not be the case.

Since every cluster in the current analysis is finite and small (i.e. the size of the cluster is finite and not close to the herd immunity threshold of the school), the data suggest that the R within schools is less than 1, but the author's analysis suggests otherwise because the analysis focuses solely on the index case transmission (e.g. not accounting for the fact that for a cluster of size 9 a single individual infects 8 individuals but each of those 8 individuals are not transmitting further). There is quite a bit of literature about stuttering chains of transmission (e.g. blumberg and lloyd-smith's work for monkeypox) that could be appropriate for addressing this issue. I believe the authors could adapt their analysis to account for multiple generations explicitly using those methods or in a different way the authors deem appropriate. This point is important because ultimately it impacts the interpretation of R, the estimation of Β for the subsequent analyses, and potentially the impact of interventions.

---

## [Author Response]

Essential revisions:1) Please reconstruct how the comparison between the US and Canadian data is done in the paper. All 3 reviewers agreed that the US data likely is hampered by overrepresentation of large super spreader events, leading to biased and inaccurate projections of the over-dispersion distribution. Please weight the 3 reviewers' suggestion for clarifying the presentation of this data.

We have restructured the paper, making the main text be exclusively about the Canadian data, and moving the US data and its analysis to the Appendix. There we highlight the many limitations of this data and explain why we may infer unrealistically large mean cluster sizes in it.

2) Please either remove the time to detection analysis or incorporate a method to account for the inherent high degree of stochasticity of this outcome.

We have removed the time to detection analysis.

3) Please define cluster more clearly.

While trying to respond to this request, we realized that there had been a misunderstanding as to how the data had originally been collected. In the previously submitted version of the paper, it was assumed that the data reflected pre-identified clusters. It turns out that the data only listed the number of cases in a school reported on each day that cases were reported. So, for example, if on May 4th, a school was reported as having 4 cases, and then on May 5th, the school reported 3 more cases, that would be included in our analysis as two separate clusters in the same school, even though they are more likely to be linked by transmission. Realizing this led us to redo our analysis. Accordingly, if cases in a school are reported within 7 days of each other, we assumed them to be in the same cluster. What we take our new clusters to be is a set of cases in one school that are linked by transmission within the school. This lead us to a different set of cluster sizes in each province, and therefore changed our parameter estimates. Generally, mean cluster size was estimated to be somewhat larger than before, with dispersion staying about the same. This did not lead to different conclusions in the rest of the paper. For example, our conclusions about what are the most likely interventions to succeed in the school setting did not change. Of course, our data only imperfectly allows us to identify true clusters, and we hope we have made the limitations of our method appropriately clear in the current manuscript.

4) Please provide a more complete discussion about the true value of q and how this might impact model conclusions.

We provide a citation to a meta-analysis of ascertainment fraction that supports our value of *q*. We discuss how a lower value of q would effect our estimates. And we still have in the SI values for our parameter with q going as low as 0.2.

Reviewer #1 (Recommendations for the authors):– Figure 1: x-axis needs labels and ticks to denote quantity.

We have fixed this.

– Please remove "not all the news is good for children": this language is too casual.

We have deleted this phrase.

– All figures: increases font size substantially please.

Done. Please let us know if this is sufficient.

– Figure 2: Please add an actual correlation plot with rho coefficients and p-values to formalize what appears to be true by eye. This will be so helpful to demonstrate that the US data is insufficient for the model used in the paper (Pardon the slang but… garbage in, garbage out). The authors can then emphasize the point that appropriately thorough data is necessary to model crowd sourced data of this nature.

Author response image 1 is a correlation plot for the Canadian data. We indicate the correlation between rate of clusters reported and incident cases in the plot. We don’t report a p-value because they are not independent data points and a standard p-value would not be informative. We would prefer to not show this plot in the paper, because the current plot are adequate for seeing the pattern in the data, that pattern is not key to any claims we are making, and the US data has since been moved to the Appendix.

**Author response image 1. sa2fig1:** 

– Table 1: given overdispersion, the median, interquartile range and full range of cluster size would be more informative than just the mean. Please add percentages to schools with multiple clusters column.

These are good suggestions. We have added fraction of schools with multiple clusters. But since most of the clusters have a single case in each province, the median is always 1, and the interquartile range is 0 or 1. Instead we have added maximum cluster size (minimum is of course 1) and index of dispersion and index of dispersion excluding the index case.

– Keeping in mind that this is a generalist journal., it would be helpful to show theoretical distributions with varying values of k as a supplemental figure. This would provide valuable context for Figure 3 and how to interpret k (in addition to Figure 4 which is great).

Thanks for this suggestion. We now have a figure showing the probability mass function for cluster size for varying parameters. We show it on a log scale for probability and for clusters only up to size 10. This makes it harder to compare to data, but otherwise it’s difficult to see the salient features of the distribution.

– The assumption that q=0.75 should be cited, or better yet, acknowledged up front as uncertain. The sensitivity analysis with varying values of q should be emphasized more when q is introduced.

We now cite a meta-analysis that includes an estimate of ascertainment in high-income countries in the Americas.

– Figure 3: would it be possible to show the median and mean on the x-axis?

We were not sure if the reviewer meant to refer to another figure. The mean is a parameter in the model (or at least is directly computed from a parameters in the model). The median is not, and is always 1 for all the model fits of the Canadian data. Apologies if we are not understanding what is being asked here.

– Figure 4: why not include all provinces in Canada?

We now show all four Canadian provinces we consider.

– Figure 5 is a bit difficult to interpret except for the most statistical minded among us, and could be a supplement (both the v and β parameters) or not included at all.

We agree that it’s difficult to interpret, which is part of the challenge of working with over dispersed distributions. But we would like to keep it in the main text, since it is one of the more useful plots we think for deciding what kinds of β values are useful in simulations.

– For all figures comparing US and Canada data, please normalize the y-axes so comparisons can be made by eye more easily.

We have moved all US data to the Appendix, since it has been pointed out that the data is probably not of high enough quality to be compared with the Canadian data.

When T < τ, how is”mingling” implemented? Does the number of contacts switch with movement from “classroom to classroom”. This needs greater explanation.

Yes, this was unclear. We have added the following sentence which we hope clarifies things: “The parameter τ can vary between some fraction of *T* (for example *T/3*, if the index case divides their time equally between three sets of n_contact_ contacts) or *T* if the set of contacts is fixed.” To answer the reviewer’s question, we don’t allow τ > T.

– Figure 7 is interesting and one of the coolest parts of the paper but lacks sufficient detail. For reduced β, increased social distancing and bubbling, the distribution is impacted to various degrees but is the total number of cases lower? Is the variance in the distributions lower? Are these differences statistically significant?

We have added comments in the text to explain more what is happening. In each of the plots, the mean number of cases is shown and have expanded the text there to make this clearer. We have also stated that all mean values are accurate to the number of digits reported, so there is no issue of statistical significance. Finally, we state the variance of the total number of infections in the figure too.

– Figure 7: The selection of R=0.5 and R=5 seems perhaps too extreme. Most persisting variants have been associated with R~1 (0.8-1.2) whereas omicron was 3-5 fold higher, albeit probably lower overall in classrooms. I would suggest at least testing an intermediate set of R values. Maybe 1 and 3?

Recall that our *R_c_* is not the same as the *R* usually used in epidemiology. Our first values of *R_c_* and k are the ones we estimated for Saskatchewan, so we think they are an appropriate choice.

For our larger value of *R_c_* we now use 2.5, which may be more realistic than or original choice of 5.

– Please mention the omicron variant in the discussion.

We have now added the following to the discussion:

“Secondly, new, more transmissible variants of SARS-CoV-2 have emerged most notably the Α variant, the Δ variant, the Omicron variant, and most recently the BA.2 strain of the Omicron variant, each with a substantially higher transmissibility than its predecessors.”

– The authors make the excellent point that the crowd sourcing method is cheaper and easier than contact tracing or phylogenetic approaches but should also mention that contact tracing would be a wonderful gold-standard to validate the accuracy of crowd sourcing in terms of identifying cluster size.

We have now added the following to the discussion:

“However, we note that if it were available, contact tracing data would be an excellent gold standard against which to check our assumptions about cluster identification.”

– Figure 8: this would be easier to interpret of the panels were square rather than rectangular.

We have made the panels closer to square.

Reviewer #2 (Recommendations for the authors):The major concern with the presented analysis is the use of sparse and potentially biased data in the analysis. The authors use compiled reports of case cluster sizes in schools across a number of jurisdictions, but have no methodological means to correct for issues clearly present in the underlying data. For example, most schools in the US only report a single cluster in the dataset, which likely biases the US dataset in a number of significant ways. This is contrasted with Canadian data where almost all schools report multiple clusters. The figures showing incident cases in schools and the community recapitulates this point with the US data showing no clear relationship between schools and the community compared with the Canadian data that shows a clear relationship. In either the case where school clusters contribute to transmission in the community or clusters are simply a byproduct of community transmission, we would expect to see correlation between the incident cases. While the authors mention these limitations to their study, I believe these data issues may present an existential issue with the current analysis as it makes it nearly impossible to interpret the results (even though I admit that the authors are fairly careful in their own framing). For example, it will be natural to wonder why Florida's Rc is nearly 8 while Texas's is around 2, and I believe there is very limited to say about the situation without controlling for underlying issues in the data between the states. I would suggest that the authors consider major methodological changes that might address the potential biases in the analysis. Perhaps limiting the analysis solely to Canadian data could also solve this issue, though there are likely biases in the dataset to consider as well.

Thank you for your comments. We agree that the quality of the data from the US is too poor to learn much from, and we have now moved it to the Appendix and explained its limitations more fully. Likewise, we have explained better how the Canadian data was collected, as well as providing some justification for our choice of ascertainment probability. Of course, one may disagree with our choice, but we hope that our sensitivity analysis can allow the reader to make their own decision about what an appropriate value of ascertainment fraction is appropriate, and make their own inferences.

The paper focuses on describing characteristics of clusters of cases in schools, and the authors give some details on the number of clusters in their datasets, however we never get a proper definition of what is considered a cluster here. According to the text on pages 11 and 12 it seems that an assumption is that all secondary cases in a cluster are infected by the same seed case (directly or indirectly). However, the interpretation of the results generally consider it to be through the direct route, even though public health policies would diverge significantly. For example a cluster that was infected outside of the school environment but were detected in school (imagine a group of close friends) would be counted in the analysis for Rc which ultimately filters to an understanding of how mitigation measures in school could control outbreaks. Alternatively, a cluster of multiple generations sputtering along might be given a large Rc when the actual R_eff for transmission might be below or close to 1.

These are indeed limitations of our analysis. We feel that we already addressed some of these limitations in the second-to-last paragraph of the discussion, but we have now added the following text as well:

“Furthermore, we assumed that all clusters consisted of an index case and a number of other cases directly infected by the index case. In reality, there may be longer chains of transmission. Any of these assumptions may bias our estimates of the distribution of *ν* and *β*.”

The cluster definition is extra important, because it also impacts the estimate of the transmission rate. The transmission rate estimation assumes all transmission is happening within schools and in a single generation. In cases where there is actually a chain of transmission, or when transmission is happening outside of the school setting the formula used to calculate β does not hold. The authors partially acknowledge this by restricting the analysis to Canadian reports, but I think it should be more explicitly discussed in the manuscript either through an attempt to handle the complexity mathematically or with a larger discussion of the interpretation of the results.

We don’t want to make our model more complicated. We could make a simulation based model, but then we lose our ability to fit (for multiple choices of q, for example) with the same ease we do here. We feel that we now discuss these limitations adequately in the discussion.

Finally, the authors use an ascertainment q=0.75, which seems high, at least for the United States reports. For the US, the CDC estimates 1 in 4 infections were reported for a period covering the authors dataset, and detection is even less likely in children who are more likely to be asymptomatic (e.g. Davies et al.). The authors mention they used an alternate model of ascertainment and running a sensitivity analysis on q, where results for q=0.5 are potentially different from the main text results. Given the very possibility that the actual value of q is in that range, I would urge that the authors revisit their main results taking this into account.

The ascertainment fraction in different regions is difficult to pin down. Indeed, there are

estimates of quite low ascertainment fractions in some regions. However, comparing the seroprevalence data from Canadian blood services to the cumulative incidence of cases, we obtained q = 1, which is admittedly unrealistically high. In the end we defer to a meta-analysis by Bobrovitz et al. In high-income jurisdictions in the Americas they report an estimated seroprevalance to cumulative incidence ratio. In the last quarter of 2020 they report 1.5 for the ratio, it decreases to 1.3 in the first quarter of 2021, and then to 1.2 in the second quarter of 2021. Taking the inverse gives numbers in the vicinity of 0.75. We added the following text:

“We base this value on the meta-analysis Bobrovitz et al. ,which reports ascertainment fractions for high-income regions in the Americas between 66% (in the last quarter of 2020) to 85% (in the second quarter of 2021).”

Reviewer #3 (Recommendations for the authors):Probably the most important observation here is that Canadian school cluster sizes and k parameters seem more consistent with a public health system that is accurately identifying (even small) school outbreaks, whereas the absence of single case outbreaks and the large fraction of very large outbreaks in the US data suggests a system which is calibrated to only identify (more obvious) superspreader events. Canada's per capita excess death rate during the pandemic has been notably lower (around 1/3) that seen in the US. Given that a relatively small contribution to mortality has come from pediatric cases, can the authors spell out a bit more explicitly how better surveillance may have blunted the pandemic's impact in Canada relative to the US?

We mainly agree with your comments, and if we had a consistent source of data from the two countries, making such a comparison would be interesting. But as the other reviewers pointed out, the difference in how the data was collected in the two countries (and even between states within the US) makes such a comparison impossible.

Accordingly, we have moved the US data and analysis to the Appendix, where we explain its limitations.

[Editors’ note: further revisions were suggested prior to acceptance, as described below.]

The manuscript has been improved and is very close to satisfactory for acceptance but there are some remaining issues that need to be addressed, as outlined below:1. Please make the abstract more balanced between methodologic gains and scientific conclusions.

Please see the following new abstract. In addition to some small changes in the beginning, we have rewritten the latter half as requested.

“The role of schools in the spread of SARS-CoV-2 is controversial, with some claiming they are an important driver of the pandemic and others arguing that transmission in schools is negligible. School cluster reports that have been collected in various jurisdictions are a source of data about transmission in schools. These reports consist of the name of a school, a date, and the number of students known to be infected. We provide a simple model for the frequency and size of clusters in this data, based on random arrivals of index cases at schools who then infect their classmates with a highly variable rate, fitting the overdispersion evident in the data. We fit our model to reports from four Canadian provinces, providing estimates of mean and dispersion for cluster size, as well as the distribution of the instantaneous transmission parameter *β*, whilst factoring in imperfect ascertainment. According to our model with parameters estimated from the data, in all four provinces (i) more than 65% of non-index cases occur in the 20% largest clusters, and (ii) reducing instantaneous transmission rate and the number of contacts a student has at any given time are effective in reducing the total number of cases, whereas strict bubbling (keeping contacts consistent over time) does not contribute much to reduce cluster sizes. We predict strict bubbling to be more valuable in scenarios with substantially higher transmission rates.”

2. Apply techniques to account for multiple generations of infection (or at least rerun the model assuming a smaller 4-5 day window to define clusters in line with the known serial interval) to ensure that this does not substantially impact the model's semi-quantitative conclusions.

We have added a section in the appendix where we recompute our parameter estimates using different windows for defining clusters: 10 days and 4 days. Changing the window never changes the estimates for the dispersion k by more than 10%. Average cluster size increases with a longer window, but by at most 11%. Decreasing the window decreases the estimate of average cluster size by at most 18%. We reference this in the main text with the following:

“We explore other choices of window in the appendix.”

and later on with

“Recall that we determined clusters by including cases in the same cluster if they were reported within 7 days of each other. In the Appendix we explore what happens if we change this window to either 4 days or 10 days. We find that estimates of κ do not change much: there is less than a 10% change in k in all cases. A window of 4 days leads to smaller cluster sizes (at most 18% smaller) and a window of 10 days leads to larger cluster sizes (at most 11% larger)”

Please see the final section of the appendix for details.

Another way we account for multiple generations of infection is in how we now estimate *β*; we use the 4 day window for *β* estimates, since that is more likely to capture only direct transmission. See our response to Reviewer #2 below.

Reviewer #2 (Recommendations for the authors):I greatly appreciate the authors revisions in response to the reviewer comments. In my previous review, I had a comment about the definition of the cluster and multiple generations of transmission. Now that the authors have provided that definition I have one major comment remaining.The authors responded to my comment with: "These are indeed limitations of our analysis. We feel that we already addressed some of these limitations in the second-to-last paragraph of the discussion, but we have now added the following text as well: 'Furthermore, we assumed that all clusters consisted of an index case and a number of other cases directly infected by the index case. In reality, there may be longer chains of transmission. Any of these assumptions may bias our estimates of the distribution of ν and β.'"Without strong evidence otherwise, I think the authors should address this concern within the analysis rather than simply a discussion. The definition of cluster now explicitly includes multiple transmission generations (through linking cases in successive weeks with one another). How many clusters have "multiple generations"? How does cluster size compare between single and multiple generation clusters? I imagine the multiple generation clusters are the largest ones, but it may not be the case.

This is a very good point. Now that we have correctly interpreted the data we have information about how clusters unfold over time in a school. Accordingly, we don’t have to only work with total cluster size, which as you point out, may include multiple generations of infection. Because other data sets only contain cluster size (such as the lower-quality American data we consider in the Appendix) we will continue to use total cluster size and the model with parameter *v* as the focus of our work. However, there is no longer a need to estimate *β* from ν; we can actually estimate the number of direct infections of an index case, and use that to estimate β. Accordingly, we estimate the number of direct infections of an index case by fitting our model to data with the cluster size defined by the 4 day window. This is more likely to capture direct infections. This leads to several minor changes in the paper, but the main difference is a modest reduction in our values of *β*. The main change in the text of the document is the following paragraph:

“Finally, we perform a second analysis using the same model, using a smaller window of time for the definition of a cluster. In this way we hope to identify only the index case and the cases directly infected by the index case. We use the model above for this (smaller) number of cases for each cluster to estimate a distribution for *ν*, but then use this in turn to estimate a distribution for the instantaneous transmission rate *β*. Our reasoning is that if *ν* is the random Poisson parameter when the index case it exposed to n people for time T, then *β* has approximately the same distribution as *ν*/(nT). Under these assumptions, *β* is also a Gamma-distributed random variable with parameter we can easily identify, from those for *ν*.”

Since every cluster in the current analysis is finite and small (i.e. the size of the cluster is finite and not close to the herd immunity threshold of the school), the data suggest that the R within schools is less than 1, but the author's analysis suggests otherwise because the analysis focuses solely on the index case transmission (e.g. not accounting for the fact that for a cluster of size 9 a single individual infects 8 individuals but each of those 8 individuals are not transmitting further). There is quite a bit of literature about stuttering chains of transmission (e.g. blumberg and lloyd-smith's work for monkeypox) that could be appropriate for addressing this issue. I believe the authors could adapt their analysis to account for multiple generations explicitly using those methods or in a different way the authors deem appropriate. This point is important because ultimately it impacts the interpretation of R, the estimation of Β for the subsequent analyses, and potentially the impact of interventions.

These are good points. Originally we were only interested in estimating the total number of cases due to the presence of one index case in a school, in part because some of our data (no longer used) only reported final cluster size. This lead us to define *R_c_*, the expected number of additional cases in the cluster, which is certainly different than the usual R. With the Canadian data we acquired later, we can actually try to estimate *R*, because there is some information about how the clusters unfold. Our estimates of *R_c_* with a cluster definition window of 4 days is probably a reasonable estimate of *R*, and hence for an estimate of *β* The suggestion to use the literature about stuttering chains of transmission for estimation in this context is an interesting one, but we have not developed it for this work.

Since using a four day window instead of a 7 day window did not change parameter estimates we added comments here and there to acknowledge the change. In addition to the changes described above, this is reflected in the main text with the following:

“For estimating the distribution of *β* we used a 4 day window for the definition of clusters, since this is more likely to include only people directly infected by the index case.”

And

“The discrepancy is even greater when we consider clusters defined by the four day window, which are even smaller.”